# Perception in Reflection

**Yana Wei** [* 1]   **Liang Zhao** [* 2]   **Kangheng Lin** [3]   **En Yu** [4]   **Yuang Peng** [5]   **Runpei Dong** [6]   **Jianjian Sun** [2]
**Haoran Wei** [2]   **Zheng Ge** [2]   **Xiangyu Zhang** [2]   **Vishal M. Patel** [1]

## Abstract

We present a *perception in reflection* paradigm designed to transcend the limitations of current large vision-language models (LVLMs), which are expected yet often fail to achieve perfect perception initially. Specifically, we propose Reflective Perception (**RePer**), a dual-model reflection mechanism that systematically alternates between policy and critic models, enables iterative refinement of visual perception. This framework is powered by Reflective Perceptual Learning (**RPL**), which reinforces intrinsic reflective capabilities through a methodically constructed visual reflection dataset and *reflective unlikelihood training*. Comprehensive experimental evaluation demonstrates RePer's quantifiable improvements in image understanding, captioning precision, and hallucination reduction. Notably, RePer achieves strong alignment between model attention patterns and human visual focus, while RPL optimizes fine-grained and free-form preference alignment. These advancements establish perception in reflection as a robust paradigm for future multimodal agents, particularly in tasks requiring complex reasoning and multi-step manipulation. Project Page: `https://weiyana.github.io/Perception-in-Reflection/`

## 1. Introduction

In advancing large vision-language models (LVLMs) (GPT-4o, 2024; Liu et al., 2024c; Bai et al., 2023), considerable attention has often been focused on enhancing the models' visual **perception** capabilities for image understanding. This emphasis stems from a fundamental assumption that

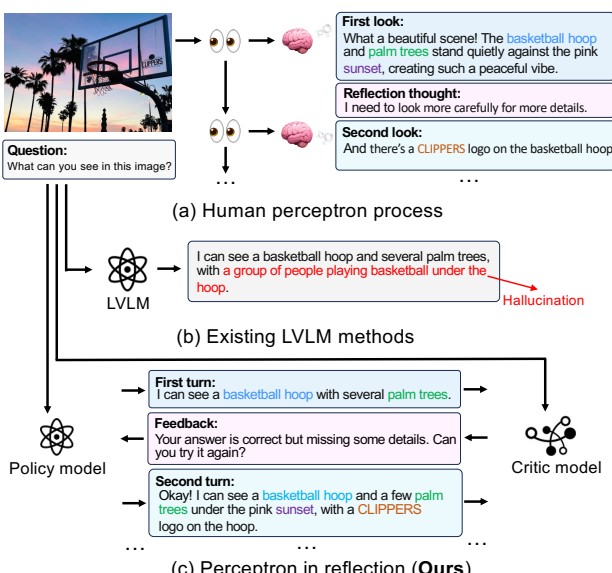

(a) Human perceptron process

(b) Existing LVLM methods

(c) Perceptron in reflection (**Ours**)

*Figure 1.* Existing LVLMs are expected to deliver accurate perceptions initially, but humans often reflect and refine answers gradually. We introduce **perception in reflection**, employing policy and critic model interactions to fully harness perceptual capabilities.

*well-trained models can achieve sufficiently accurate initial perception.* Such perceptual accuracy enables the model to process visual inputs and generate appropriate responses in a *single pass* (Liu et al., 2024c;b; Wang et al., 2024). However, the frequent occurrence of hallucinations and misperceptions hinders their wider applicability in real-world scenarios. As shown in Figure 1, even for simple scenes, models may generate hallucinatory descriptions (e.g., as seen in (b)) or fail to capture essential details (e.g., as observed in the initial response in (c)). This raises an important consideration: *Are current perception paradigms inherently limited, or might there be a more reasonable paradigm?*

Some methods (Chen et al., 2023; Liu et al., 2024d; Yu et al., 2023) attempt to mitigate this through a sort of visual chain-of-thought (CoT) (Wei et al., 2022) reasoning. They establish a paradigm that first executes fine-grained perceptual tasks (such as grounding object locations (Chen et al., 2023; Shao et al., 2024), structures (Liu et al., 2024d) or identities (Yu et al., 2023)) before engaging in broader

*Equal contribution  [1]Johns Hopkins University  [2]StepFun [3]BUPT  [4]HUST  [5]Tsinghua University  [6]UIUC. Correspondence to: Yana Wei <ywei66@jh.edu>, Vishal M. Patel <vpatel36@jhu.edu>.

*Proceedings of the 42nd International Conference on Machine Learning*, Vancouver, Canada. PMLR 267, 2025. Copyright 2025 by the author(s).

perception. However, these approaches face a key limitation: the reliance on specialized tasks and data formats that are difficult to generalize across all vision-language tasks, *e.g.*, box CoT can not be used in math geometry problems, making it challenging to achieve consistent visual perception across diverse scenarios. Furthermore, CoT does not change the original single-pass manner. When perceptual errors occur, it is unable to adjust and rectify them.

Shifting the view to the real world, we can observe that humans, as shown in Figure 1, typically do not perceive in a single step, rather, they establish cognition through gradual observation. This iterative process enables humans to continually enrich, refine, and enhance their perceptual outcomes. Drawing inspiration from this, we think that *a reasonable perception paradigm for LVLMs should be iterative rather than a single-pass.* In other words, the ability to reflect and improve over multiple rounds is not just a desirable feature; it's a *fundamental requirement* for LVLMs to achieve robust and generalizable perception.

In this paper, we propose a novel perceptual mechanism, termed **Re**flective **Per**ception (**RePer**). Its purpose is to enable LVLMs to, like humans, use a perception-feedback loop to gradually establish precise visual cognition. To achieve this, we make RePer a dual-model architecture, *i.e.*, *policy* model and *critic* model, to enable LVLMs to conduct *percption* and *reflection* separately in terms of multi-turn dialogues between policy and critic model. In this way, LVLMs distill lessons from past experiences, gradually direct attention toward critical visual primitives, and thereby produce more accurate and refined responses.

Although LVLMs inherently possess reflective perception capabilities, this ability is instable and has not been effectively activated (Kumar et al., 2024). To this end, we further propose a **R**eflective **P**erceptual **L**earning (**RPL**) approach. Through strategic temperature sampling and a hybrid evaluation system combining model and rule-based rewarding, we construct an online, multi-turn *visual reflection dataset*. This dataset exhibits progressive improvements in both perception accuracy and response quality across dialogue turns. Building upon this, we propose *reflective unlikelihood training*, an imitation learning approach (Ross et al., 2011; Swamy et al., 2023) that calibrates the model's preferences across responses of varying quality, thereby mitigating *behavioral collapse* (Kumar et al., 2024) where models tend to generate suboptimal responses in early turns.

Extensive experiments demonstrate that RePer achieves superior performance across various benchmarks including image understanding, hallucination detection and detailed image caption, *e.g.*, $54\%$ CAPTURE on DetailCaps (Dong et al., 2024a) and $51\%$ accuracy on HallusionBench (Guan et al., 2024). Using GPT-4o (GPT-4o, 2024) and DALLE-3 (Betker et al., 2023), we validate its enhanced perception

capabilities from both discriminative and generative perspectives. Comprehensive ablation studies on data construction, training strategies, reflection rounds, and critic designs verify RePer's generalizability, establishing it as a fundamental paradigm for advancing multimodal perception.

In order to thoroughly unveil the underlying mechanisms behind *perception in reflection*, we further conducted a series of analytical experiments. Our comprehensive experimental analysis reveals two key findings:

- RePer can effectively *migrate image attention towards human-aligned regions* through iterative refinement. This implies that the perceptual pattern utilized by RePer aligns more closely with that of humans.

- RPL can be regarded as a *free-form preference optimization* framework that unifies various preference learning paradigms, *e.g.*, DPO (Rafailov et al., 2024), and LiPO (Liu et al., 2024e), while enabling fine-grained supervision through explicit feedback signals.

These two key findings underscore the crucial value of *perception in reflection* in enhancing multimodal understanding and reasoning capabilities. We believe it will become an essential capability for multimodal agents in the future, particularly in complex visual reasoning (Xie et al., 2024; Małkiński & Mańdziuk, 2022) and multi-step manipulation (Sampat et al., 2022; Kim et al., 2024) tasks.

## 2. Perception in Reflection

In this section, we first define our problem and formalize the objective from a reinforcement learning perspective (Section 2.1). We then elaborate on how models learn to perceive through reflection, encompassing both data construction and training strategies (Section 2.2). Finally, we present the inference algorithm for reflective perception during deployment (Section 2.3).

### 2.1. Problem Definition and Formulation

**Perception in LVLMs.** Perception, as a concept in the field of computer vision (He et al., 2016; Ren et al., 2016; He et al., 2017), refers to the process of interpreting and understanding sensory, *ie.,* vision, information from the environment. In the context of LVLM, we typically define perception as the process by which the model recognizes and understands the image or video. The perception capability of the model will directly determine the accuracy of its understanding and reasoning towards real world.

**Perception in Reflection.** Our goal is to mimic human perception, establishing a perceive-feedback loop through LVLM's iterative attempts to enhance image comprehension and response accuracy. In pursuit of this, we model our

challenge through the lens of reinforcement learning (RL), inspired by SCoRe (Kumar et al., 2024) and RISE (Qu et al., 2024). To be specific, given a dataset $\mathcal{D} = \{(I_i, x_i, y_i^*)\}_{i=1}^N$ of images $I_i$, questions $x_i$, and oracle responses $y_i^*$, we aim to train an LVLM policy $\pi_\theta(\cdot \mid [I, x, \hat{y}_{1:t}, f_{1:t}])$. This model, given an image $I$ and question $x$, along with $t$ previous attempts $\hat{y}_{1:t}$ and feedback prompts $f_{1:t}$, is designed to perceive the image as accurate as possible and deliver the most correct possible answer $y$. Formally, given a verifier $r(y, y^*)$ to assess the correctness of model response $y$ compared to oracle answer $y^*$, we aim to derive a policy that utilizes the aforementioned information to produce the outputs with the highest correctness reward over $T$ rounds:

$$\max_{\pi_\theta} \sum_{t=1}^{T} \mathbb{E}_{I,x,y^* \sim \mathcal{D}, \hat{y}_t \sim \pi_\theta(\cdot|[I,x,\hat{y}_{1:t-1},f_{1:t-1}])} r(\hat{y}_t, y^*).$$

(1)

Section 2.1 resembles a multi-round Markov Decision Process (MDP) (Qu et al., 2024) or can be viewed as an RL or supervised finetune (SFT) objective. It is noteworthy that every historical attempt is synchronously optimized to maximize the ultimate reward.

## 2.2. Reflective Perceptual Learning

Despite existing LVLMs often possessing intrinsic self-reflection capabilities (Liu et al., 2024a), these abilities have been shown to be remarkably fragile (Kumar et al., 2024). In other words, they struggle to adaptively refine their responses based on given feedback (as shown in Figure 12). To address this limitation, we propose Reflective Perception Learning (RPL), a methodology that trains models to continuously enhance their previous responses through imitation learning (Ross et al., 2011; Swamy et al., 2023). We first elaborate on the data collection and training objective.

**Data Construction.** Naturally, we structure a multi-turn dialogue encompassing the sequence of posing questions, providing responses, receiving erroneous feedback, and subsequently re-responding and re-evaluating. This iterative process is designed to cultivate and demonstrate reflective perception capabilities within the trained models.

Practically, we expect the model to, **(1)** generate *diverse* responses based on all past answers and feedback, thereby enabling the *exploration* during reflection towards a perception with fewer errors; **(2)** gradually produce more *accurate* answers in multi-turn dialogues, ensuring the *convergence* of the reflective chain. To meet these requirements, we construct a visual reflection dataset for model imitation. Figure 2 gives an overview, with detailed steps as follows:

***Step-1*: Initial Candidate generation.** We employ *temperature sampling* to generate diverse candidate answers per image-question pair. This approach ensures sufficient variation in response style, detail level, and accuracy while

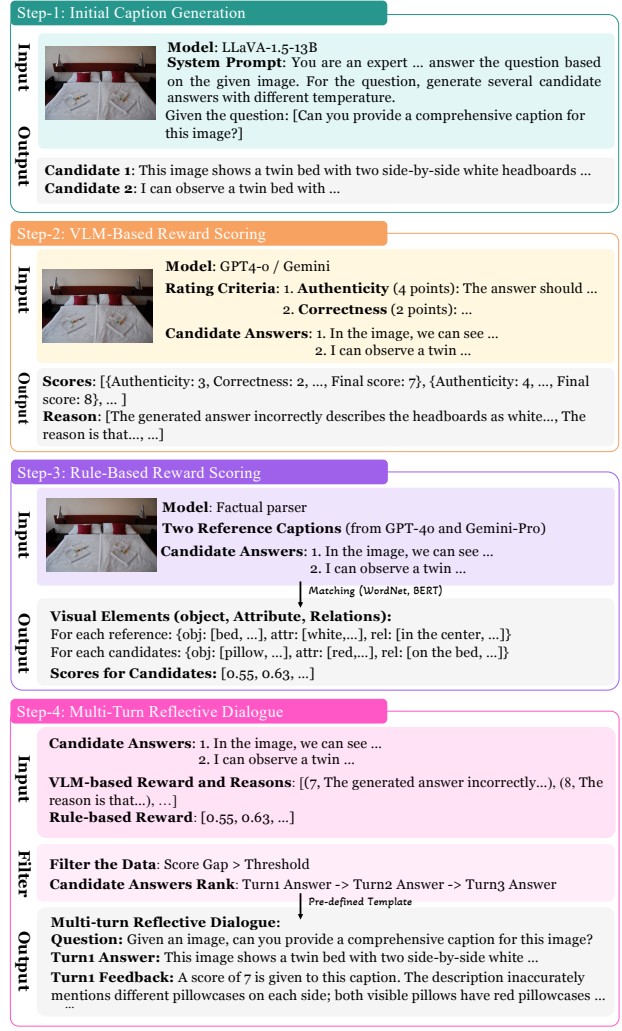

*Figure 2.* **Data construction pipeline of visual reflection dataset.**

maintaining semantic relevance.

***Step-2*** **VLM-Based Reward Scoring.** For the generated multiple candidate responses, we employ a robust Visual-Language Model (VLM) to conduct a comprehensive and multifaceted evaluation, yielding fine-grained scores.

***Step-3*** **Rule-Based Reward Scoring.** Then we design a pipeline to extract key elements, *e.g.*, objects, attributes, and relations, from both images and responses, and establish matching rules to compute alignment scores.

***Step-4*** **Reflective Dialogue construction.** After obtaining the candidate answers and their corresponding reward scores, we select samples meeting two criteria: (a) a minimum score gap between the highest and lowest responses, and (b) at least one response scoring above the specified points. Then the filtered responses are structured into $N$ rounds of reflective dialogue, progressing from lowest to highest scores. To this end, we curate a dataset

$\tilde{\mathcal{D}} = \{\{(I_t^i, x_t^i, \tilde{y}_t^i, f_t^i, r_t^i)\}_{t=1}^T\}_{i=1}^N$, where $\tilde{y}_t^i$ is sampled from model outputs, $f_t^i$ represents specific feedback, and $r_t^i$ denotes the corresponding reward score.

---

**Algorithm 1** Reflective Perception (RePer)

---

1: Initialize Policy, Critic model: $\pi_\theta, r_\theta$
2: Generate initial perception response $y_0$ using $\pi_\theta$ given image $I$ and language instruction $x$
3: Generate initial evaluation $r_0, f_0$ using $r_\theta$ given $(I, x, y_0)$
4: Set $t \leftarrow 0$
5: **while** $t <$ max trials **do**
6:     Generate perception response $y_t$ using $\pi_\theta$ given $(I, x, y_0, r_0, f_0, ..., y_{t-1}, r_{t-1}, f_{t-1})$
7:     Generate evaluation $r_t, f_t$ using $r_\theta$ given $(I, x, y_0, r_0, f_0, ..., y_{t-1}, r_{t-1}, f_{t-1}, y_t)$
8:     Increment $t$
9: **end while**
10: **return**

---

Two points merit attention. First, it is crucial to reward answer of each round using a hybrid scoring mechanism. This approach aims to align the model with both *rule-based* and *model-based* reward systems (Mu et al., 2024), thereby maximizing its ability to generalize to complex real-world scenarios. Second, we aim to devise responses based on the self-generated outputs of the model, thereby facilitating an *online* optimization process. This is intended to minimize the risk of the model *overfitting* to non-reflective capabilities (Kumar et al., 2024; Qu et al., 2024; Tang et al., 2024).

**Reflective Unlikelihood Training.** Based on the constructed data, we apply imitation learning (Ross et al., 2011; Swamy et al., 2023) to simulate reflective perception. This learning process necessitates the disregard of textual patterns, focusing instead on the cultivation of capabilities.

More critically, we seek to prevent the model from overfitting to multi-turn responses and avoid the *behavioral collapse* (Kumar et al., 2024) where the model consistently generates *suboptimal* initial replies. In previous efforts, both RISE (Qu et al., 2024) and SCoRe (Kumar et al., 2024) primarily utilized SFT for imitation learning. However, RISE employed the exponent of *centered rewards* to mitigate this issue, while SCoRe utilized *reward shaping* to counteract. In this paper, we propose a method that simultaneously balances likelihood and unlikelihood (Welleck et al., 2019), formalized as follows:

$$\max_\theta \mathbb{E}_{\circ^i \sim \tilde{D}} \sum_{t=1}^T \sigma_t^i \log \pi_\theta(\tilde{y}_t^i | \circ_t^i) + \alpha(1 - \sigma_t^i) \log(1 - \pi_\theta(\tilde{y}_t^i | \circ_t^i)), \quad (2)$$

where $\circ$ denotes a single sampling instance from our constructed dataset $\tilde{\mathcal{D}}$, and $\sigma_t^i = F(r_t^i)$ represents the normalization of reward $r_t^i$. $\alpha$ is a constant term that adjusts the

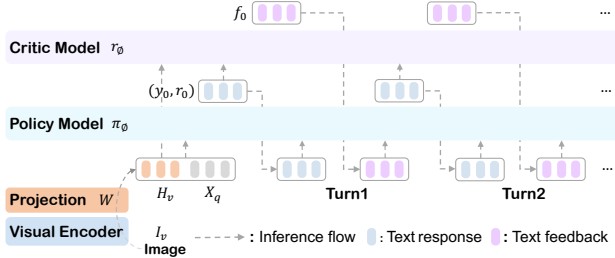

*Figure 3.* **Inference pipeline of reflective perception**.

unlikelihood loss scale.

Essentially, we employ rewards to balance likelihood and unlikelihood. In the initial rounds where the reward is lower (smaller loss weight), there is a predisposition towards unlikelihood, promoting the penalization of the response. Conversely, in subsequent rounds where the reward is higher (larger loss weight), there is a tendency towards likelihood, encouraging rewarding of the response. We will elaborate this on Section 3.2.

### 2.3. Reflective Perception

As shown in Algorithm 1 and Figure 3, we define reflective perception during the inference process as a *collaborative interaction* between the well-trained policy and critic agents.

Initially, the policy model observes and provides its perceptual results, which are then evaluated by the critic model. The critic model assesses the policy's perception, providing both a score to gauge the quality of the perception and a rationale for the score given. Following this, the policy model reflects on its perceptual errors, informed by the critic's evaluations on self-generated responses from previous rounds, and produce a new perceptual response. The critic model then delivers updated feedback based on all previous perceptions and critiques. This iterative process continues until a predetermined limit is reached.

Through an iterative reflective perception mechanism, the model can achieve human-like visual perception patterns, as empirically validated by the experimental results demonstrated in Figure 4.

## 3. Discussion

### 3.1. RePer Progressively Aligns Human Attention.

The reflective capabilities of language models have been widely studied (Liu et al., 2024a; Kumar et al., 2024; Qu et al., 2024). It is suggested that these models inherently possess a reflective ability that, although unstable, can improve the certainty of outputs and activate deeper levels of knowledge (Liu et al., 2024a). Yet, the effectiveness of this capacity in LVLMs remains an unexplored question.

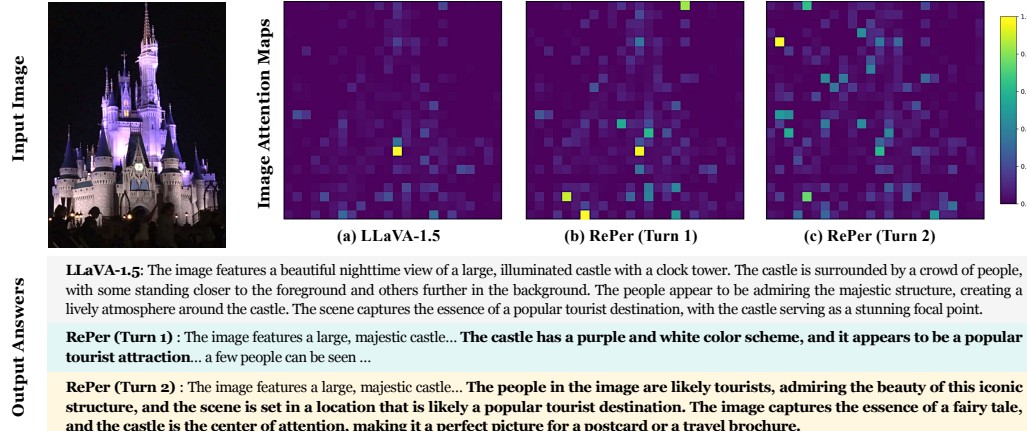

*Figure 4.* **Comparison of image attention maps between LLaVA-1.5 and RePer**, highlighting RePer's broader activation of image tokens and its ability to generate more detailed and accurate answers. While LLaVA-1.5 over-focuses on "people", RePer correctly attends to the main subject, "castle," progressively activating more relevant tokens for improved perception.

To investigate, we begin by visualizing the detailed image attention of the model for each round of the model's responses. As shown in Figure 4, as the rounds progress, the model *accurately shifts its attention emphasis* towards the correct image tokens, rather than over-focusing on a few insignificant ones. Hence correspondingly, the hallucinations and erroneous perceptions initially displayed by the model gradually decrease. Essentially, this represents a ***progressive alignment towards ground-truth human attention***.

Figure 6a provides further quantitative support, showing a log-linear increase in average image token activations over five rounds of reflection. Specifically, we calculated the average image token activations across 1,000 cases to measure how the model's attention to images varies during reflective perception. This is important because responses with fewer hallucinations are associated with higher average activations of image tokens (Huang et al., 2024). Our findings suggest that visual reflection *gradually unlocks the model's inherent visual capabilities*, focusing attention on salient image context and progressively mitigating hallucination.

### 3.2. RPL is a Free-Form Preference Optimization.

Revisiting the data construction in RPL, we essentially transform *listwise preference data* with precise feedback and scores into multi-turn dialogues grading from poor to good quality. This prompts the inquiry: is RPL fundamentally a preference optimization process?

Revisiting Equation (2), for a given sample ∘ and its $T$ dialogue iterations, the objective is articulated as follows:

$$L^i = \underbrace{\sigma_1 \log \pi_\theta(\tilde{y}_1|\circ_1)}_{\text{less likelihood}} + \underbrace{\alpha(1 - \sigma_1) \log(1 - \pi_\theta(\tilde{y}_1|\circ_1))}_{\text{more unlikelihood}} + ... +$$
$$\underbrace{\sigma_T \log \pi_\theta(\tilde{y}_T|\circ_T)}_{\text{more likelihood}} + \underbrace{\alpha(1 - \sigma_T) \log(1 - \pi_\theta(\tilde{y}_T|\circ_T))}_{\text{less unlikelihood}}.$$

(3)

As aforementioned, to develop reflective perception capabilities, we create multi-turn data that progresses from poor to good responses, with rewards increasing linearly from rounds $1$ to $T$. As a result, in the initial rounds, the model mainly penalizes poor samples (*more unlikelihood*), while in later rounds, it gradually shifts to rewarding good samples (*more likelihood*). This helps the model avoid overfitting to poor initial samples and, importantly, allows it to progressively learn to distinguish between good and bad samples.

From another perspective, we can view RPL as a form of *reward modeling*. Unlike popular LLM-based reward modeling methods such as DPO (Rafailov et al., 2024) and LiPO (Liu et al., 2024e), RPL does not propagate gradients to the remaining negative samples. Yet, back-propagation over multi-round dialogues is actually not isolated. With each response *contextualizing* all previous responses, as denoted by $\circ_t = [I, x, \hat{y}_{1:t-1}, f_{1:t-1}]$, each sample implicitly establishes a ***partial increasing preference order***.

Moreover, it is worth noting that RPL holds a significant advantage over previous reward modeling approaches: flexibility in handling diverse preference samples—*pairwise or listwise, scalar or fine-grained feedback*-based rewards—while maintaining stable training. Additionally, the use of detailed feedback aids error highlighting, facilitating object-level or even token-level preference that direct optimization more precisely. Our analyses in Section 4.6 further confirms this.

## 4. Experiments

### 4.1. Implemental Details

**Datasets.** To construct the training dataset as illustrated in Section 2.2, we begin by randomly sampling 10,000 images from the LLaVA-665K (Liu et al., 2024c) dataset. For each image, we prompt the model to generate 8 different

*Table 1.* **Model Performance Comparison of RePer with Baselines and State-of-the-Art Models.** RePer outperforms across six benchmarks, with the best results highlighted in **bold**.

| Model | MMHal-Bench | | HallusionBench | | | Detailcaps-4870 | | | LLaVABench | GAVIE | | GAPE | | |
|---|---|---|---|---|---|---|---|---|---|---|---|---|---|---|
| | Score ↑ | Hal rate ↓ | aAcc ↑ | fAcc ↑ | qAcc ↑ | CAPTURE ↑ | Precision ↑ | Recall ↑ | | Relevancy ↑ | Accuracy ↑ | Authen. ↑ | Correct. ↑ | Total ↑ |
| MiniGPT-4 7B | - | - | 35.78 | 10.12 | 8.79 | - | - | - | 45.1 | - | - | - | - | - |
| mPLUG-Owl 7B | - | - | 43.93 | 10.40 | 9.45 | - | - | - | - | - | - | - | - | - |
| InstructBLIP 7B | - | - | 45.26 | 10.11 | 9.45 | 51.81 | 65.22 | 45.01 | 59.8 | - | - | - | - | - |
| LLaVA-SFT+ 7B | 1.88 | 0.68 | 33.65 | 8.96 | 5.93 | 51.13 | 64.38 | 44.28 | 44.6 | 6.68 | 4.85 | 27.62 | 12.47 | 70.09 |
| LLaVA-RLHF 7B | 1.67 | 0.76 | 31.23 | 14.16 | 7.69 | 52.21 | 63.61 | 45.93 | 44.9 | 4.88 | 4.27 | 27.93 | 12.64 | 70.68 |
| VOLCANO 7B | 2.06 | 0.62 | 26.50 | 10.69 | 6.37 | 50.88 | 66.23 | 43.35 | 54.0 | 7.12 | 5.35 | 31.63 | 14.52 | 78.78 |
| LLaVA-SFT+ 13B | 1.92 | 0.65 | 46.37 | 22.25 | 18.24 | 51.08 | 64.48 | 44.04 | 55.8 | 6.85 | 5.20 | 30.00 | 13.44 | 74.88 |
| LLaVA-RLHF 13B | 2.09 | 0.69 | 36.20 | 15.32 | 14.73 | 52.05 | 64.56 | 45.35 | 62.6 | 4.66 | 4.33 | 30.06 | 13.59 | 75.36 |
| VOLCANO 13B | 2.15 | 0.64 | 40.69 | 19.36 | 13.40 | 51.21 | 66.47 | 43.65 | 66.0 | 7.55 | 5.59 | 31.34 | 14.32 | 78.17 |
| LLaVA-1.5 7B | 2.02 | 0.61 | 35.65 | 17.92 | 11.21 | 51.03 | **67.27** | 42.19 | 60.2 | 6.50 | 5.28 | 30.19 | 13.58 | 75.16 |
| **+RePer** | 2.51 | 0.53 | 38.70 | 19.65 | 14.29 | 52.89 | 66.81 | 45.69 | 60.7 | 6.91 | 6.04 | 33.16 | 14.94 | 80.88 |
| LLaVA-1.5 13B | 2.35 | 0.58 | 43.85 | 20.81 | 14.95 | 51.23 | 66.26 | 43.77 | 66.95 | 6.65 | 5.49 | 31.27 | 14.12 | 77.37 |
| **+RePer** | **2.61** | **0.52** | **51.00** | **22.83** | **20.00** | **54.73** | 64.74 | **49.1** | **67.6** | **7.67** | **6.86** | **34.11** | **15.33** | **82.54** |

captions sampled with temperatures ranging from 0.0 to 1.4 in increments of 0.2. To filter high-quality samples, we retain instances from VLM-based scoring where the highest score exceeds 9 and the score disparity (difference between the highest and lowest scores) is greater than 4. Similarly, for rule-based scoring, we retain cases with a highest score above 0.55 and a score disparity exceeding 0.2. Using the generated captions, rewards, and templates from Figure 2, we create the **visual reflection dataset**, containing 11,065 samples from 8,101 images. These samples are distributed as follows: 3,649 for one conversation turn, 2,621 for two turns, and 3,795 for three turns.

**Models Training and Inference.** Our experiments are based on the LLaVA-1.5 (Liu et al., 2024b) architecture. We directly supervised finetune the instruct model on our generated datasets. All models are trained for one epoch on 8 NVIDIA A100 GPUs with a batch size of 8 and a learning rate of 1e-6. Only the parameters of the LLM module are fine-tuned, while the rest remain frozen. In reflective unlikelihood training (Equation (2)), rewards are normalized to [0,1] by dividing with their maximum values ($F$), serving as likelihood weight ($\sigma$). The constant term $\alpha$ is set as 10.0. During the inference stage mentioned in Section 2.3, we defaultly use LLaVA-Critic (Xiong et al., 2024) as the critic model.

### 4.2. Main Results

To evaluate the visual perception capabilities of RePer, we conducted assessments across five widely-used benchmarks, covering a range of tasks: image understanding (LLaVABench (Liu et al., 2024c)), hallucination detection (HallusionBench (Guan et al., 2024), MMHal-Bench (Sun et al., 2023b), GAVIE (Liu et al., 2023a)), and detailed image captioning (DetailCaps (Dong et al., 2024a)). As shown in Table 1, we compared RePer not only with classic state-of-the-art multimodal baselines including MiniGPT-4 (Zhu et al., 2023), mPLUG-Owl (Ye et al., 2023), InstructBLIP (Dai et al., 2024), LLaVA (Liu et al., 2024c),

LLaVA-RLHF (Sun et al., 2023a), LLaVA-1.5 (Liu et al., 2024b) but also with Volcano (Lee et al., 2023), a multimodal model trained with self-feedback guided refinement.

As shown in Table 1, RePer consistently outperforms baseline models across benchmarks and model scales. Its notable improvement on DetailCaps (+3.64% in 7B and +6.83% in 13B) highlights its ability to generate more accurate and detailed captions through multi-turn refinement and RPL. The increased recall rate (+8.30% in 7B and +12.17% in 13B) for visual elements demonstrates RePer's enhanced perception of details. This results in consistent improvements on general and hallucination-related benchmarks, reducing hallucinations without sacrificing image understanding.

### 4.3. GPT-4o-Assisted Perception Evaluation (GAPE)

We introduce GPT-4o-Assisted Perception Evaluation (GAPE) to simulate human-like perception assessment. Designed to complement traditional closed-set image captioning benchmarks (Chen et al., 2015; Agrawal et al., 2019), GAPE evaluates model-generated captions by leveraging human-aligned prompts with GPT-4o (Peng et al., 2024) without the need for human-annotated groundtruth answers. Specifically, given an image and a prompt, GPT-4o evaluates the generated captions across five dimensions: *Authenticity*, *Correctness*, *Detail*, *Coherence*, and *Completeness*. The evaluation prompts align with the "Rating Criteria" outlined in Figure 8. To better highlight differences in caption quality, these dimensions are scored on a larger scale from 0 to 100, offering a human-like and nuanced assessment of caption performance.

**Results** As shown in Table 1 and Table 6, our RePer consistently outperforms other methods, demonstrating its effectiveness in enhancing model's perceptual capabilities. Notably, we observe the most significant improvement in *Authenticity*, which evaluates the model's tendency to hallucinate non-existent objects. This substantial gain can be attributed to our unlikelihood training objective, which ef-

Table 2. Comparison of human preference metrics and GAPE scores for 13B-scale models.

| Model | Mean Rank ↓ | Top-1 ↑ | GAPE ↑ |
|---|---|---|---|
| LLaVA-1.5 | 2.46 | 15.13% | 77.37 |
| Volcano | 2.01 | 35.29% | 78.17 |
| **LLaVA-1.5+RePer** | **1.53** | **63.87%** | **82.54** |

Table 3. Image captioning comparison on 13B models using the CLIP-Image-Score metric and its variants with DINO/DINOv2 as Image encoders.

| Model | CLIP | DINO | DINOv2 |
|---|---|---|---|
| LLaVA-1.5 | 67.43 | 40.56 | 41.02 |
| +RePer | **67.85** | **42.19** | **42.12** |

fectively penalizes misaligned visual descriptions.

**Evaluation Reliability** To assess the reliability of our automatic evaluation, we conducted a human study comparing caption preferences across three 13B-scale models (LLaVA-1.5, Volcano, and RePer) on 119 samples from the GAPE benchmark. For each image, model-generated captions were anonymized, shuffled, and ranked by six expert annotators, following the same five rating criteria used in GAPE. We report two metrics: 1) *Mean Rank* — the average position of each model across all rankings; and 2) *Top-1 Rate* — the percentage of times a model's caption was ranked highest by humans. As shown in Table 2, the human preference rankings align closely with the GPT-4o-based GAPE scores, validating the reliability of our automatic evaluation. Notably, RePer achieves the top rank in 63.87% of cases, demonstrating strong agreement with human judgment.

### 4.4. Evaluation via Text-to-Image Reconstruction

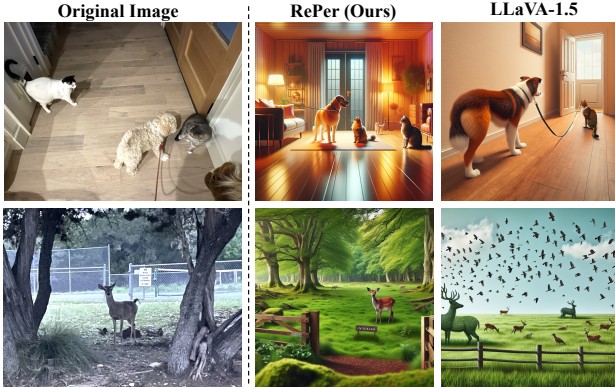

*Figure 5.* We use DALLE-3 (Betker et al., 2023) as a text-to-image model to reconstruct images using generated captions. Compared to the original image, reconstructed images from LLaVA-1.5 (Liu et al., 2024b) captions lack key objects or include extraneous ones, indicating incomplete descriptions or hallucinations.

We assess image captioning performance, a key perceptual

application, using the CLIP-Image-Score metric from VisualFactChecker (Ge et al., 2024). This metric evaluates caption accuracy and detail by comparing the similarity between an original image and its text-to-image generated version (DALLE-3 (Betker et al., 2023)), using the caption as a prompt. By comparing the raw and reconstructed images, the metric detects hallucination-related discrepancies, providing a unique perspective on caption quality. To enhance this evaluation, we substitute the CLIP model with DINO (Caron et al., 2021) and DINOv2 (Darcet et al., 2023) for a more thorough assessment.

As shown in Table 3, our RePer consistently outperforms the baselines, underscoring the superior quality of its captions. Figure 5 presents visual examples of the reconstruction process. In the second example, LLaVA 1.5 falsely mentions, "There are several birds scattered throughout the scene," exhibiting hallucination. In contrast, the caption from our RePer produces a reconstructed image closely resembling the original, demonstrating its superior accuracy and ability to avoid hallucinations.

### 4.5. Ablation Studies

**Reflection Turns** We analyze the impact of reflection turns on model performance using LLaVA-Critic and GPT-4o as the critic. As shown in Figure 6b, increasing reflection turns improves performance on the DetailCaps-4870 benchmark, reducing hallucinations and enhancing detail perception. This aligns with our attention analysis (Figure 4), suggesting that iterative reflection helps the model better focus on relevant image regions.

**Scoring Disparity for Data Construction** We also examine the effect of scoring thresholds in data selection (Section 2.2) on DetailCaps and HallusionBench. As shown in Figure 6c, optimal performance is achieved with samples having highest scores above 9 and score disparities of at least 4, indicating that high scoring disparity helps select challenging yet high-quality training samples.

**Unlikelihood Loss** We further study the influence of unlikelihood loss weight $\alpha$ (from Equation (2)) on reducing behavior collapse in initial responses using DetailCaps and HallusionBench. As shown in Figure 6d, a weight of 10.0 achieves optimal performance by effectively balancing the penalization of undesirable responses while preserving valuable content.

### 4.6. Further Analysis

**Critic matters, RPL matters more.** To assess RPL and different critics' impact on RePer, we compare its performance with and without RPL, using critics LLaVA-Critic (Xiong et al., 2024) and GPT-4o (GPT-4o, 2024), across multiple

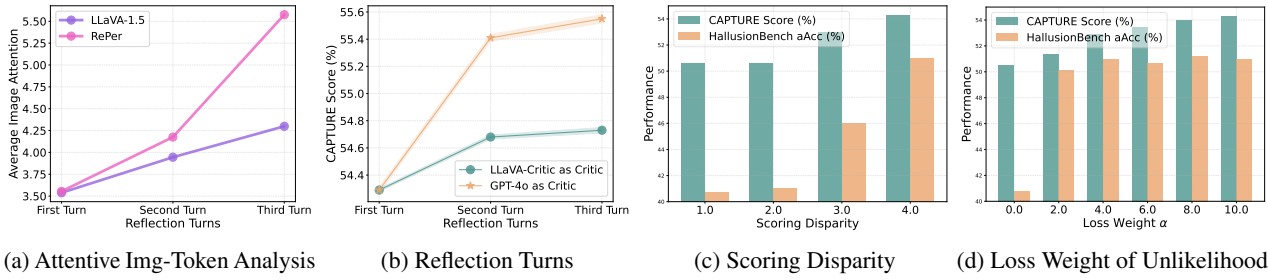

(a) Attentive Img-Token Analysis   (b) Reflection Turns   (c) Scoring Disparity   (d) Loss Weight of Unlikelihood

*Figure 6.* (a) Increase in activated average image attention across reflection turns. (b-d) Ablation studies.

*Table 4.* Comparison of RePer and RePer without RPL under varying critics and reflection turns on Detailcaps-4870.

| Critic | Turn | RePer | RePer w.o. RPL |
|---|---|---|---|
| GPT-4o (GPT-4o, 2024) | 1 | 54.29 | 51.22 |
| | 2 | 55.41 | 52.28 |
| | 3 | 55.55 | 53.9 |
| LLaVA-Critic (Xiong et al., 2024) | 1 | 54.29 | 51.22 |
| | 2 | 54.68 | 52.25 |
| | 3 | 54.73 | 53.85 |

*Table 5.* RPL vs. Preference Optimization Methods.

| Method | DetailCaps | HallusionB | GAVIE | LLaVAB |
|---|---|---|---|---|
| LLaVA-1.5-13B | 51.22 | 24.43 | 5.65 | 66.95 |
| +DPO (Rafailov et al., 2024) | 50.53 | 25.61 | 5.28 | 66.2 |
| +LiPO (Liu et al., 2024e) | 52.31 | 25.04 | 6.27 | **69.5** |
| +RPL | **54.73** | **31.28** | **6.86** | 67.6 |

reflection turns on DetailCaps. In Table 4, GPT-4o yields superior results due to its strong generative and discriminative abilities, while LLaVA-Critic also shows consistent improvements, indicating RePer's adaptability to different critics. Even without RPL, RePer benefits from reflection; however, RPL further amplifies this effect, leading to a stronger initial-turn performance and demonstrating the effectiveness of the imitation learning approach. For a detailed view of RePer's iterative reflection process, refer to Figure 7.

**RPL is essentially fine-grained preference optimization.** As detailed in Section 3.2, RPL's imitation learning in reflective dialogues can be seen as listwise preference optimization with detailed feedback and explicit rewards. We compare it to similar methods: DPO, which optimizes Bradley-Terry (Bradley & Terry, 1952) using preference pairs with the largest score differences, and LiPO, which optimizes learning-to-rank (Liu et al., 2009) using all preference data ranked by reward. Specifically, for each multi-turn correction trajectory ranked by reward, DPO selects only the highest- and lowest-scoring responses to form a single preference pair, while LiPO constructs all possible response pairs, treating the higher-scoring response in each as preferred. Table 5 shows RPL's clear advantages, especially in caption and hallucination metrics. We speculate this success stems from: 1) fine-grained critic feedback that facilitates effective corrections, lacking in DPO/LiPO; and 2) unlikelihood training without KL constraints, which helps

counteract multimodal hallucinations.

## 5. Related Work

The remarkable scaling laws (Kaplan et al., 2020) of LLMs (Touvron et al., 2023a; Xu et al., 2024) in terms of parameters and data have driven the advancement of LVLMs. BLIP-2 (Li et al., 2023a) pioneered the use of Q-Former to bridge visual encoders with large language models, explicitly supervising the vision-language alignment while autoregressively generating vision-related text. Works like LLaVA (Liu et al., 2024c;b), MiniGPT-4 (Zhu et al., 2023), and Qwen-VL (Bai et al., 2023; Wang et al., 2024) have demonstrated the sufficiency of text autoregression for visual understanding and have progressively simplified the vision-language connector using techniques such as cross-attention (Ye et al., 2023), linear layers (Liu et al., 2024c; Zhao et al., 2023), MLPs (Liu et al., 2024b; Zhang et al., 2024; Dong et al., 2024b), and convolutions (Yu et al., 2023; Wang et al., 2024), all while maintaining consistent performance.

Despite relentless scaling of visual encoders (Tong et al., 2024a; Wei et al., 2024a), language decoders (Wang et al., 2024), and visual-textual corpora (Li et al., 2024; Wei et al., 2024b), LVLMs have yet to achieve a qualitative leap in perceptual acuity or hallucination mitigation. Some approaches attribute hallucinations to visual (Tong et al., 2024b) or linguistic biases (Li et al., 2023b), seeking to counter them through online (Liu et al., 2023b) or offline (Leng et al., 2024) corrections. Others (Yu et al., 2024; Zhu et al., 2024; 2025) take a more direct route, modulating the model's visual attention preferences by aligning with human judgment via Reinforcement Learning from Human Feedback (RLHF) (Ouyang et al., 2022). Yet, disappointingly, these efforts have failed to tackle the root issue: models still reflexively respond to perceptual challenges, regardless of their complexity.

LLMs often use step-by-step reasoning (Wei et al., 2022) to avoid giving premature answers. However, this linear process can falter with complex problems, leading to factual inaccuracies and hallucinations (Miao et al., 2023).

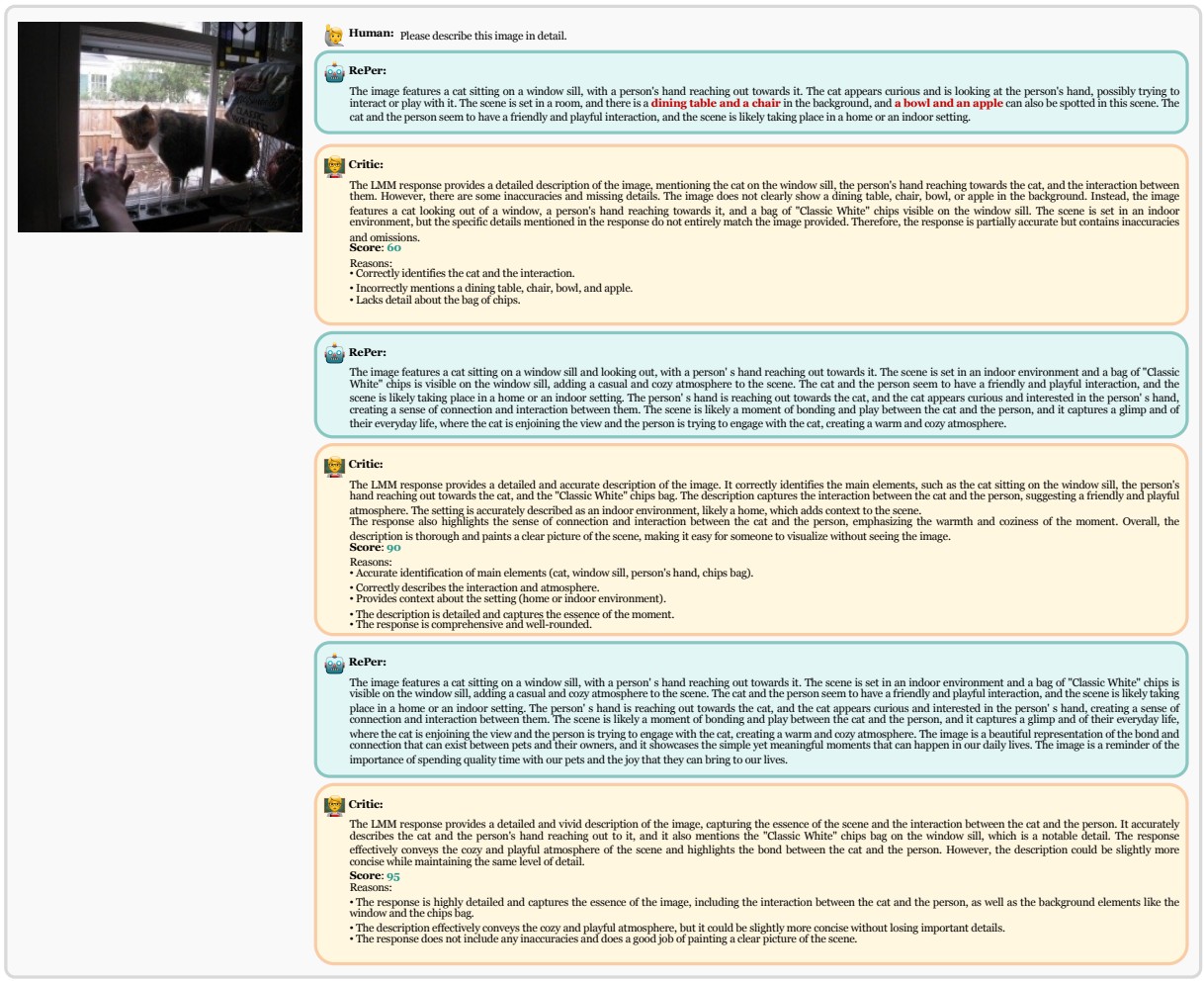

*Figure 7.* **Multi-turn Refinement via Critic Feedback.** Initially, RePer generates hallucinated objects but corrects them in the second turn based on critic feedback. By the third turn, it enhances the description with more details, improving a response that already scored 90. The increasing feedback scores highlight RePer's iterative refinement and progressive improvement.

To counter this, some approaches use external feedback to guide reasoning (Shinn et al., 2024; Yao et al., 2022), while others harness the model's reflective abilities for self-correction (Liu et al., 2024a; Miao et al., 2023; Qu et al., 2024; Kumar et al., 2024). These methods employ an iterative "answer-reflect-reanswer" loop, significantly improving performance on complex challenges.

Some LVLMs require preliminary image parsing tasks like grounding (Chen et al., 2023; Shao et al., 2024), parsing (Liu et al., 2024d; Wei et al., 2024c; Chen et al., 2024), or identification (Yu et al., 2023; 2025) before responding. While this chain-of-thought-style approach moderately improves performance, other methods (Cao et al., 2024; Wu & Xie, 2024) focus on locating relevant image regions and cropping them to assist with fine-grained perception. However, these methods often struggle with complex scenarios and may increase hallucination. Recent work explores iterative refinement using internal (Liu et al., 2024a; Lee et al., 2023)

or external (Liao et al., 2024) rewards. Despite promising results, these approaches lack systematic training frameworks and do not sufficiently explore the underlying principles of their mechanisms. We address these limitations by proposing RePer and RPL, with comprehensive theoretical and empirical analysis.

## 6. Conclusion

Perception in reflection addresses a key limitation in current LVLMs: the unrealistic expectation of perfect initial responses. Instead, it provides a robust fallback mechanism, empowering the model to adjust and converge on the correct answer even when initial predictions fall short. Powered by reflective perceptual learning, we create a system that can generalize more effectively across varied and complex visual scenarios, ensuring that the model is not only accurate but also resilient and adaptive in real-world applications.

## Impact Statement

This paper proposes a model that leverages reflective perception and structured preferences to improve image understanding, captioning precision, and reduce hallucinations in AI-generated text—a known concern in multimodal systems. As the dataset and training are based on model-generated outputs and curated feedback, we believe the work poses minimal immediate societal risk. Nonetheless, responsible deployment should account for fairness, bias, and potential misuse.

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

# Appendix

In this appendix, we provide additional details to complement the main paper. Specifically, Appendix A elaborates on the Visual Reflection Dataset, while Appendix B further validates RePer's alignment with human visual focus. Appendix C presents details of the proposed GAPE benchmark along with its complete results. Finally, Appendix D showcases additional examples illustrating the strong capabilities of RePer.

## A. Construction Details of Visual Reflection Dataset

This section provides additional details on the data construction process introduced in Section 2.2 and Section 4.1.

### A.1. Step-1: Initial Candidate Generation

To generate diverse responses, we employ *temperature sampling*, producing eight candidate captions per image across different temperature values, ranging from $0.0$ to $1.4$ in increments of $0.2$. Higher temperatures generally lead to lower response quality, often introducing hallucinated objects or less precise descriptions.

### A.2. Step-2: VLM-Based Reward Scoring

We define evaluation criteria for high-quality image captions, which guide the reward scoring process through carefully designed prompts (as shown in Figure 9). The reward score ranges from 0 to 10 and assesses five key aspects:

- Authenticity: Whether the caption contains hallucinated objects.

- Correctness: Whether all described attributes and relationships are factually correct.

- Detailness: Whether the description is sufficiently detailed, covering all relevant attributes of objects.

- Coherence: Whether the caption is logically consistent, without contradictions.

- Completeness: Whether the caption comprehensively covers all relevant aspects of the image, including both foreground and background elements.

### A.3. Step-3: Rule-Based Reward Scoring.

Inspired by (Dong et al., 2024a), we design rule-based rewards to quantify the alignment between image elements and textual descriptions. This evaluates visual preference through a structured pipeline:

**Reference Caption Generation**    We prompt strong VLMs (GPT-4o and Gemini-Pro) using *"Please describe this image in detail."* to generate reference captions for each image.

**Element Extraction**    We extract objects, attributes, and relations from both reference captions and candidate answers using Factual Parser (Li et al., 2023c), while applying stop-word filtering to remove irrelevant terms. To filter irrelevant elements, a stop word list is curated for abstract nouns (e.g., "foreground", "background") that do not correspond to image content. LLaMA2-13B-chat (Touvron et al., 2023b) and Factual Parser are used to extract candidate nouns from ShareGPT4V-102k (Chen et al., 2025). Words recalled by Factual Parser but missing in LLaMA2-13B-chat are reviewed, and high-frequency terms are validated by human experts. This process results in the final stop word list.

**Elements Matching**    We implement a three-stage matching strategy to evaluate visual elements:

- Exact Matching: Directly aligns identical objects, attributes, and relations.

- Synonym Matching: Uses WordNet (Miller, 1995) to identify synonym sets and assigns a 1.0 match score for synonymous elements.

- Soft Matching: Applies BERT (Devlin et al., 2018) to compute cosine similarity between embeddings of unmatched elements, selecting the highest similarity score per element.

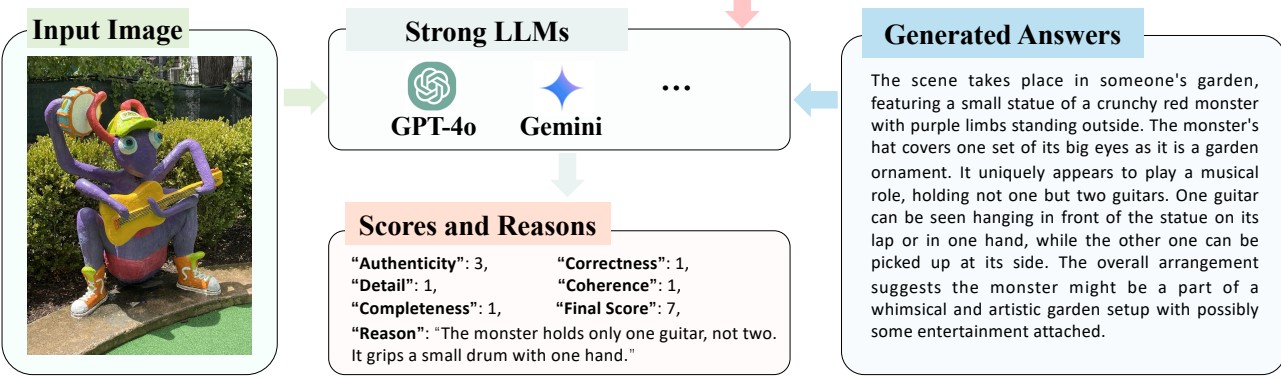

**Rating Criteria**

As a senior computer vision and NLP researcher, and an impartial data quality assessor, please evaluate the quality of the generated caption in relation to the given image on a scale of 0 to 10. Each aspect has an upper bound which has been marked as follows:

1. **Authenticity** (4 points): The answer should not describe an object or aspect that doesn't exist in the image.
2. **Correctness** (2 points): Does the answer correctly identify all the objects present in the image?
3. **Detail** (2 points): How well does the answer describe each object and its attributes?
4. **Coherence** (1 points): Is the answer logically consistent and understandable, maintaining a steady flow throughout?
5. **Completeness** (1 points): Does the answer cover all aspects in the image?

Give lower points if the answer fails in any of these aspects. A score of 10 indicates that the generated answer is accurate, detailed, coherent, complete and does not hallucinate any details not present in the image. The final score for the answer is the sum of scores in each aspect. Please also offer your reason for this score, and output the dict as the answer.

**Input Image**

**Strong LLMs**

GPT-4o    Gemini    ...

**Generated Answers**

The scene takes place in someone's garden, featuring a small statue of a crunchy red monster with purple limbs standing outside. The monster's hat covers one set of its big eyes as it is a garden ornament. It uniquely appears to play a musical role, holding not one but two guitars. One guitar can be seen hanging in front of the statue on its lap or in one hand, while the other one can be picked up at its side. The overall arrangement suggests the monster might be a part of a whimsical and artistic garden setup with possibly some entertainment attached.

**Scores and Reasons**

**"Authenticity"**: 3,      **"Correctness"**: 1,
**"Detail"**: 1,      **"Coherence"**: 1,
**"Completeness"**: 1,      **"Final Score"**: 7,
**"Reason"**: "The monster holds only one guitar, not two. It grips a small drum with one hand."

*Figure 8.* **Pipeline for Generating VLM-Based Reward Scores.** Given an image and a generated answer, a predefined rating criterion for high-quality captions is formulated as a prompt. A strong LLM then evaluates the response, producing a reward score along with a rationale for the assigned score.

**Final Matching Score** The final Rule-based reward aggregates scores from all three stages to compute precision, recall, and F1-score. The final caption quality score is a weighted sum of the three F1 scores, with default weights of Object:Attribute:Relation = 5:2:2.

### A.4. Step-4: Reflective Dialogue Construction

To construct reflective dialogues, we first apply data filtering based on the criteria outlined in Section 2.2 (Step-4) and Section 4.1 (Datasets). We then organize responses into a structured reflection-driven dialogue format, using the template illustrated in Figure 8.

For VLM-based rewards, given an image, its candidate answers, and scores with reasons from GPT-4o, we sort candidate answers from lowest to highest score. The multi-turn dialogue is constructed as follows:

- In the first turn, the model is given the initial question.

- In subsequent turns, each reflective prompt includes the score, explanation, and a question asking the model to improve its response.

- We construct dialogues of 1-3 turns, ensuring that the final ground-truth answer is always the highest-scoring candidate.

- For multi-turn cases, the first-turn response is always the lowest-scoring candidate, enabling a progressive refinement process.

This answer-critic iterative refinement encourages the model to learn from mistakes, gradually correcting its responses over multiple turns. For the rule-based rewards, there is no reasoning provided, and the scoring criteria differ from those of VLM-based rewards, which leads to a different interpretation of the relative score differences. Therefore, each round's prompt is selected from a predefined prompt pool that expresses the meaning of *"could you generate a better answer."*

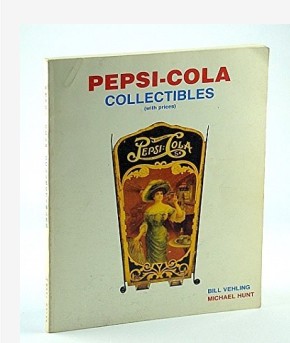

*Question:* Please generate a detailed caption that accurately describes the image.

*Response:* In the image, we can see that an older Pepsi bottle or vase collectible is sitting high above other, lesser-valued objects. Additionally, a collectible Pepsi-Co bottle with a yellow back stand and satchel is on display at ground level.\n\nThe image shows a collection of antique or valuable Pepsi-Collectibles arranged, which might belong to an avid collector of Pepsi Co. memorabilia, emphasizing the historical value and distinct features of the antiques.

*Critic Score and Rationale:* This caption can only get the score **5**. The description hallucinates objects that don't exist in the image, such as a bottle or vase and the arrangement of multiple items. The image actually is of a book titled 'Pepsi-Cola Collectibles,' showing one illustrated collectible Pepsi sign on the cover. The text about the scene with various collectibles is inaccurate. Could you generate a better one?

*Response:* The image features a book titled \"Pepsi-Cola Collectibles\" by Michael Hunt, which showcases old and rare Pepsi-Cola items collected from various sources. The book is a guide for collectors, likely providing information about the history and value of these items.\n\nThe front cover of the book features an illustration of a Pepsi-Cola display cabinet, which reflects the nostalgic atmosphere of the collection. The book is likely filled with fascinating illustrations and information about vintage Pepsi-Cola items, making it an exciting read for collectors and enthusiasts of Pepsi-Cola memorabilia.

*Critic Score and Rationale:* This caption qualifies for a score of **9**. The generated answer accurately describes the title and general subject of the book. However, the authorship attribution is partially incorrect, as the image shows 'Bill Vehling' and 'Michael Hunt'. The description of the book's content and cover illustration is appropriate but lacks detailed specifics. Please suggest an improved one.

*Response:* The image features a book titled \"Pepsi-Cola Collectibles\" by Bill Vehling and Michael Hunt. The book is displayed on a white background, with the cover prominently visible. The book appears to be a guide or catalog for collectors of Pepsi-Cola memorabilia, showcasing various items and their values.

*Figure 9.* **Illustration of data construction based on the designed template.** The **responses** are structured progressively from the lowest to the highest score. Each **reflective prompt** consists of three components: **reward score**, **rationale** for the score, and a follow-up question requesting an improved response. This iterative refinement process guides the model towards generating more accurate and detailed captions.

*Table 6.* Comparison of RePer's Performance with Baselines and State-of-the-Art Models on the GAPE Benchmark.

| Model | GAPE | | | | | |
|---|---|---|---|---|---|---|
| | Authenticity ↑ | Correctness ↑ | Detail ↑ | Coherence ↑ | Completeness ↑ | Total ↑ |
| LLaVA-SFT+ 7B | 27.62 | 12.47 | 12.27 | 9.61 | 8.11 | 70.09 |
| LLaVA-RLHF 7B | 27.93 | 12.64 | 12.44 | 9.55 | 8.11 | 70.68 |
| VOLCANO 7B | 31.63 | 14.52 | 13.89 | 9.86 | 8.90 | 78.78 |
| LLaVA-SFT+ 13B | 30.00 | 13.44 | 13.09 | 9.76 | 8.58 | 74.88 |
| LLaVA-RLHF 13B | 30.06 | 13.59 | 13.39 | 9.71 | 8.61 | 75.36 |
| VOLCANO 13B | 31.34 | 14.32 | 13.76 | **9.85** | 8.9 | 78.17 |
| LLaVA-1.5 7B | 30.19 | 13.58 | 13.15 | 9.78 | 8.46 | 75.16 |
| **+RePer** | 33.16 | 14.95 | 13.95 | 9.87 | 8.96 | 80.88 |
| LLaVA-1.5 13B | 31.27 | 14.12 | 13.48 | 9.81 | 8.69 | 77.37 |
| **+RePer** | **34.11** | **15.33** | **14.26** | 9.70 | **9.15** | **82.54** |

# B. Alignment with Human Focus

To further support our claim in Section 3.1 that RePer's image attention better aligns with human visual focus, we conducted a targeted human evaluation. We recruited six annotators (PhD/Master's students in computer vision and NLP) and randomly sampled 100 images from the test set. For each image, we collected image attention maps generated by RePer and LLaVA-1.5 during inference. These maps were anonymized and randomly shuffled before being shown to annotators, who were asked: "Which attention map better reflects your own visual focus if you were to answer this question?" The annotation interface is shown in Figure 10. We report the win rate, defined as the percentage of cases where RePer's image attention map was preferred over LLaVA-1.5's. RePer was favored in 70.27% of cases, indicating stronger alignment with human visual perception.

# C. Details of GAPE

As introduced in Section 4.3, GAPE employs evaluation prompts aligned with the "Rating Criteria" outlined in Figure 8. The evaluation score ranges from 0 to 100, with the following weight distribution: Authenticity (40 points), Correctness (20 points), Detail (20 points), Coherence (10 points), and Completeness (10 points).

GAPE serves as a complement to traditional image captioning benchmarks. Existing benchmarks, such as COCO Cap-

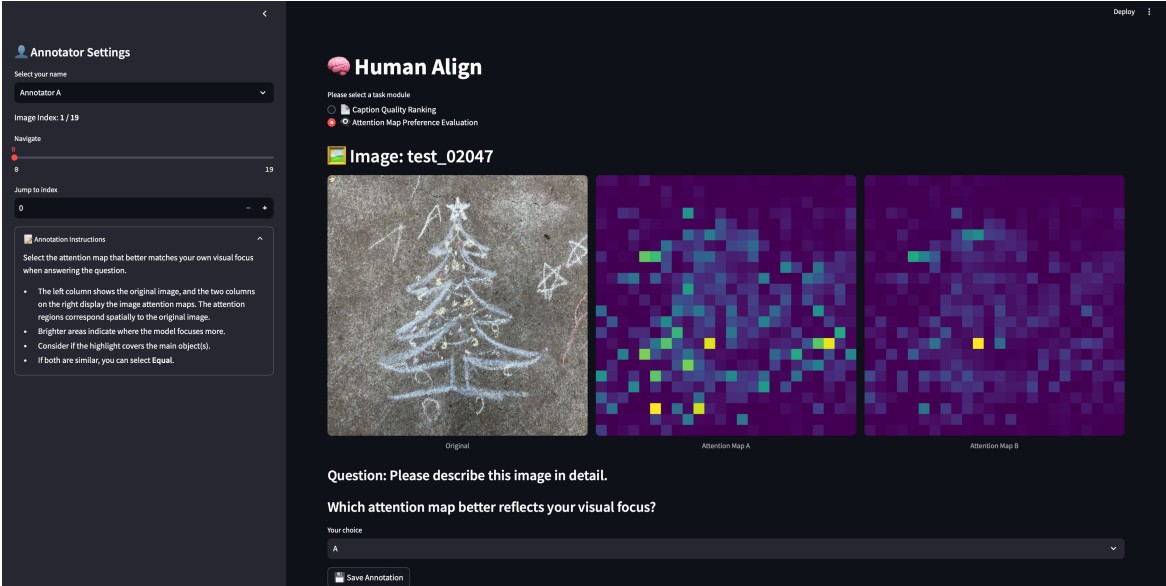

*Figure 10.* **Human annotation interface for assessing attention map alignment with human focus.** For each image, annotators selected the attention map that best matched their own viewing behavior.

tion (Chen et al., 2015) and NoCaps (Agrawal et al., 2019), rely on language metrics like BLEU, ROUGE, and CIDEr, which are constrained by closed-set vocabularies and require human-annotated ground truth captions, making them less scalable for evaluating vision-language models (VLMs) that generate diverse descriptions. In contrast, GAPE provides a more flexible and robust evaluation pipeline, operating without human-annotated ground truth and leveraging LLM-based assessment to better align with human judgment, while accommodating the open-ended nature of caption generation.

Table 6 presents a detailed comparison of model performance on GAPE across all evaluation aspects.

Figure 11 shows the interface of human annotation mentioned in Section 4.3.

## D. Case Study

Figure 12 illustrates a failure case of the baseline model LLaVA-1.5 (Liu et al., 2024b) in a multi-turn conversation with a critic model (LLaVA-Critic (Xiong et al., 2024)). In this setup, LLaVA-1.5 first generates an initial response and then attempts to refine its answer based on feedback from the critic. Despite receiving detailed and accurate feedback, the model fails to correct its mistakes in the second turn, continuing to include hallucinated objects (highlighted in red) from the initial response. Furthermore, in the third turn, it introduces additional hallucinated objects, demonstrating its inability to properly leverage feedback for refinement. This failure also results in stagnant or even lower scores assigned by the critic. This underscores the necessity of equipping models with stronger feedback-driven self-correction mechanisms to improve response accuracy and mitigate hallucinations.

Figure 7 illustrates a successful refinement case of RePer. While the initial response contains hallucinated objects, RePer corrects these errors in the second turn based on feedback from the critic model. By the third turn, it further enhances the description by adding more details to an already accurate response, which had received a score of 90 from the critic. Throughout the multi-turn conversation, the feedback score steadily increases, reflecting the model's progressive improvement. This demonstrates RePer's strong ability to iteratively refine its answers based on prior responses and feedback, effectively guiding itself toward a more detailed and accurate final answer.

Figure 13 compares the initial-round outputs of the baseline model and RePer, highlighting RePer's strong ability to generate accurate and detailed descriptions. While the baseline model frequently introduces hallucinated objects, RePer consistently produces more faithful image descriptions. Additionally, we present images generated by DALLE-3 (Betker et al., 2023) as part of the evaluation process in Section 4.4. The higher-quality captions from RePer lead to synthesized images that more closely resemble the original inputs, further demonstrating its effectiveness as a good captioner.

Figure 14 presents question-answering cases that assess the model's image understanding capabilities. The responses are generated in a single-turn answer format without iterative refinement.

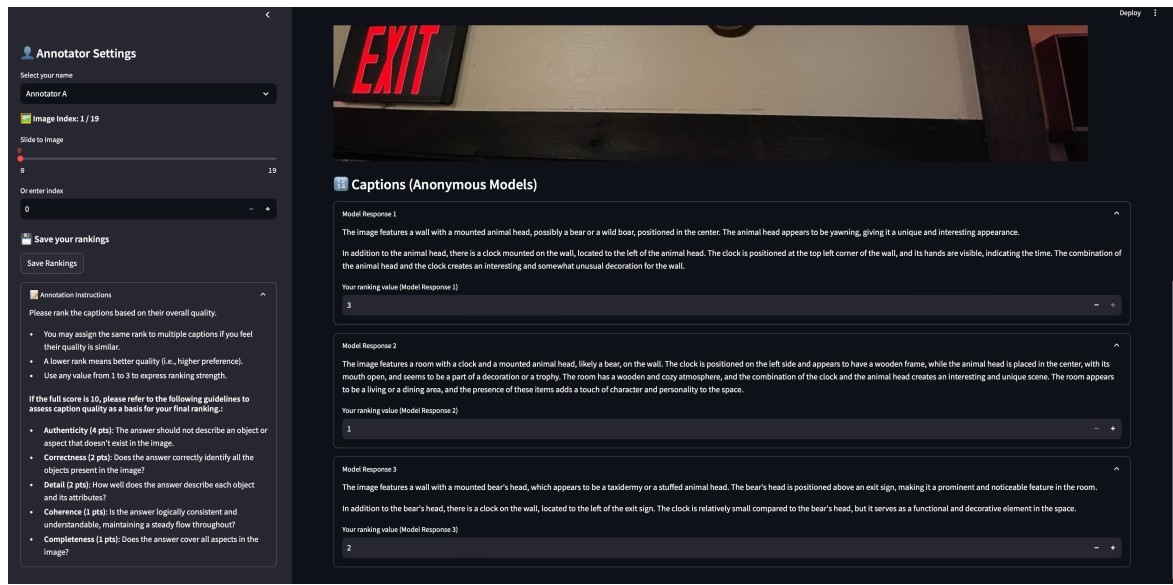

*Figure 11.* **Human annotation interface for caption quality assessment.** Annotators ranked anonymized captions from three models per image using five quality criteria (authenticity, correctness, detail, coherence, completeness).

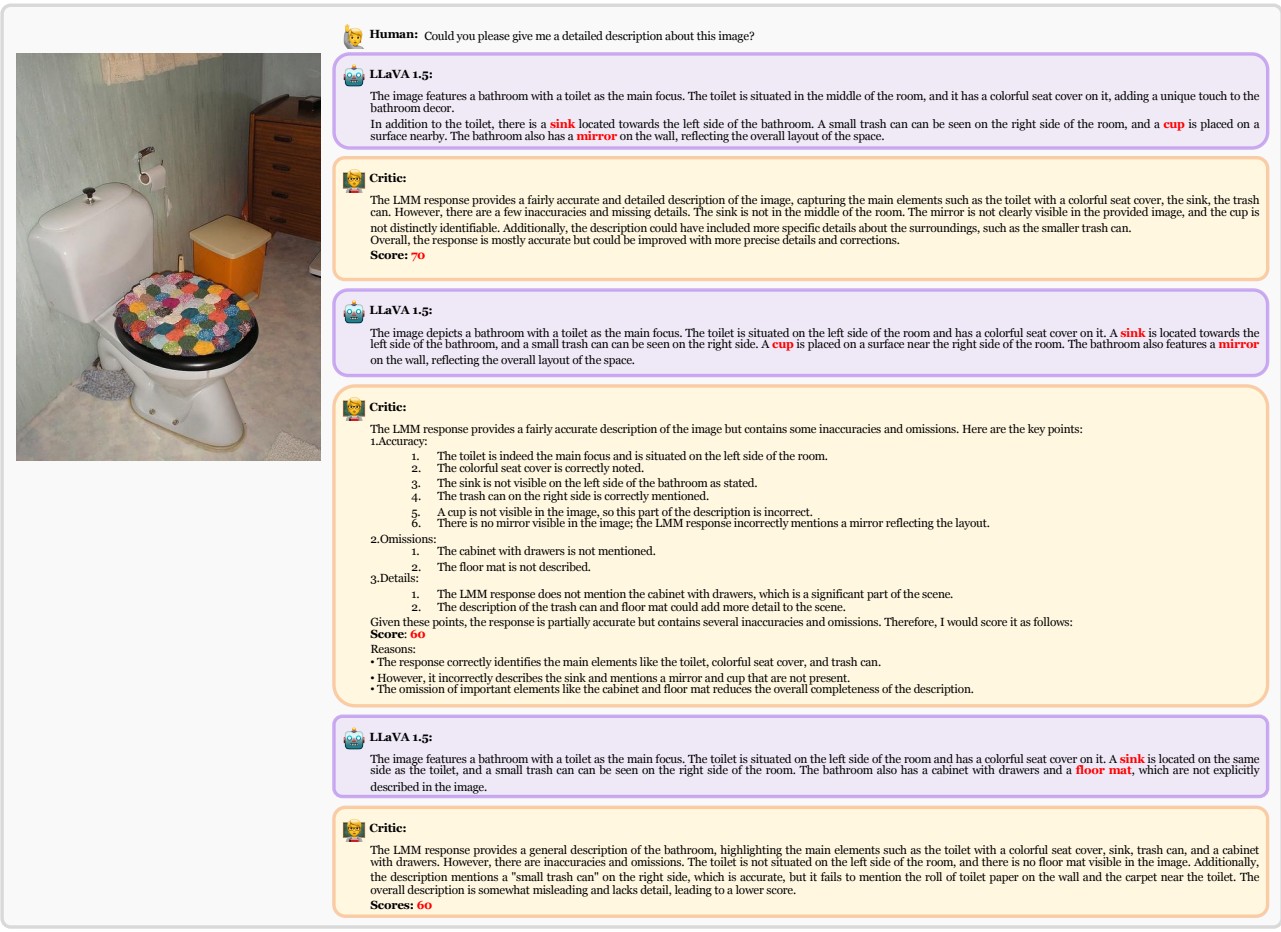

*Figure 12.* **Failure case of LLaVA-1.5 in answer refinement.** Despite receiving critic feedback on the initial response, LLaVA-1.5 fails to correct the hallucinated objects (marked in red) in the second turn and further introduces the nonexistent "floor mat" in the third turn.

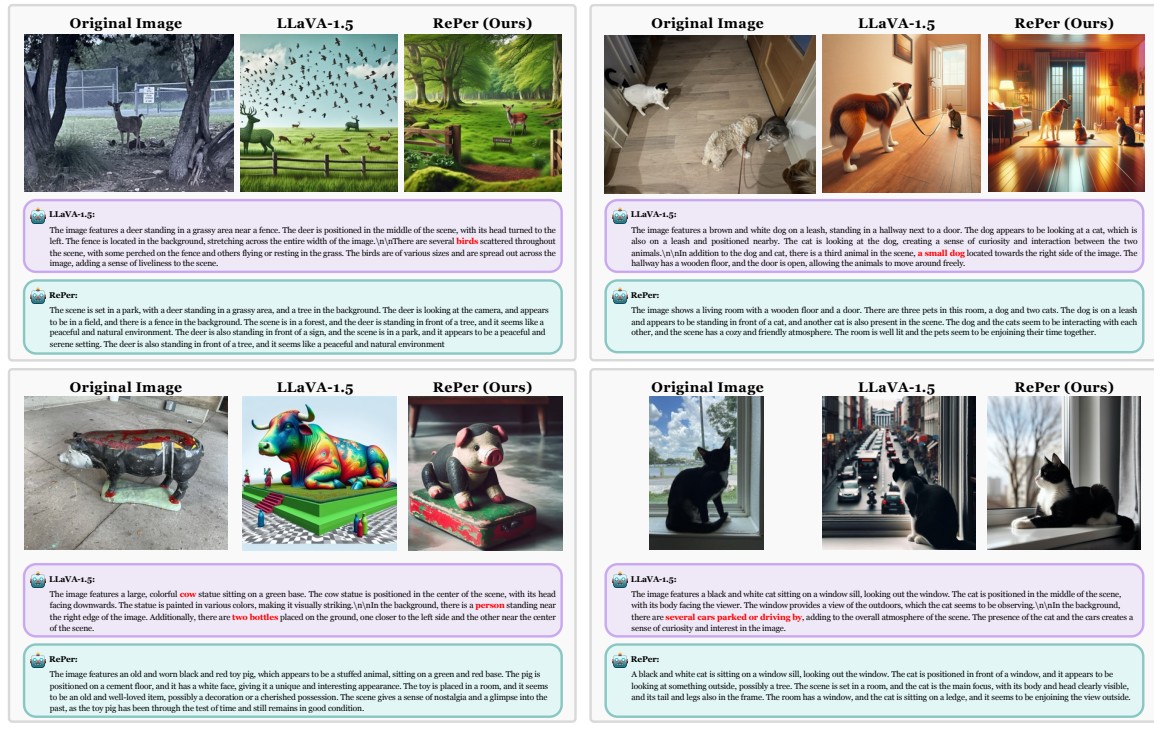

*Figure 13.* **Comparison of initial-round outputed image captions between the baseline model and RePer.** RePer generates more accurate and detailed descriptions, while the baseline model often introduces hallucinated objects. The corresponding DALLE-3 generated images further illustrate that RePer's captions lead to more faithful image reconstructions.

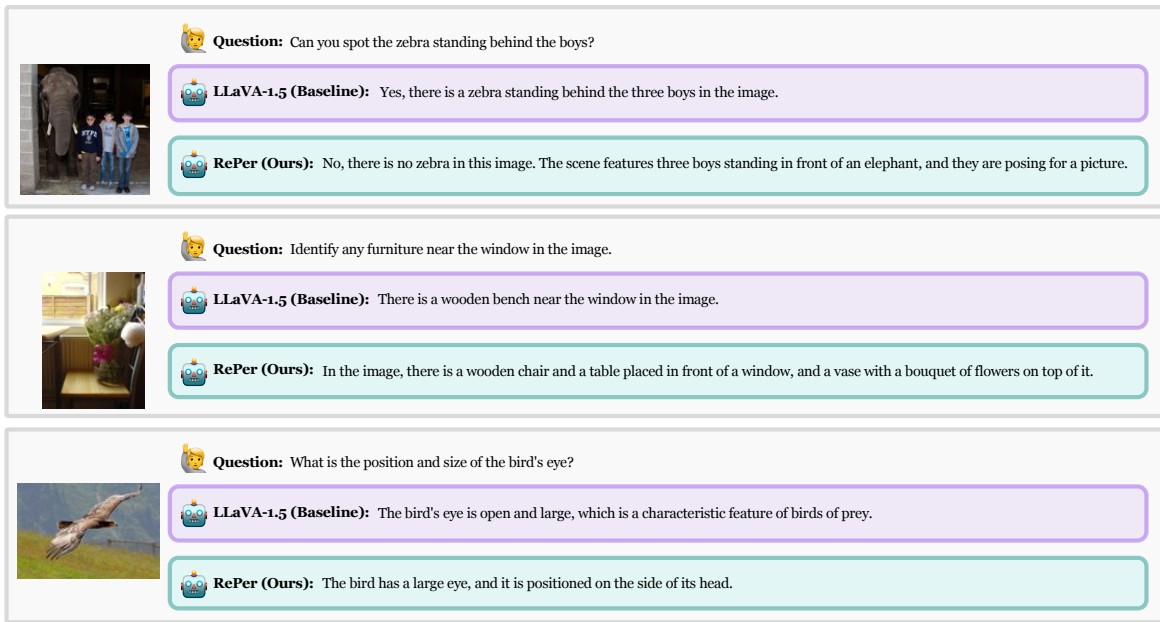

*Figure 14.* **Question-answering cases evaluating models' image understanding.** Responses are generated in a single-turn format without iterative refinement.

