# OpenReview forum: "Perception in Reflection"
_ICML.cc/2025/Conference — ICML 2025 poster_

### Official Review · Reviewer_SX8u · 2025-03-13

**Overall Recommendation:** 3

**Summary:**

This paper introduces Reflective Perception (RePer), a system for improving VLMs through iterative self-reflection. It adopts a policy–critic framework, where a policy model generates outputs, and a critic model provides feedback to refine responses over multiple turns. The paper also proposes Reflective Perceptual Learning (RPL), a training strategy leveraging automatically generated reflection data and unlikelihood loss to improve factual accuracy and reduce hallucinations. Experiment results show RePer's quantifiable improvements in image understanding, captioning precision, and hallucination reduction

**Claims And Evidence:**

Most of the claims make sense to me; however, there are two worth discussing.

1. "RePer achieves strong alignment between model attention patterns and human visual focus".
From the discussion in Section 3.1, I can see the model shifts its attention after learning, but it is unclear to me how the model aligns with human perception.

2. Advantages in complex reasoning tasks.
The paper emphasizes RePer's advantages in complex reasoning tasks, but the experimental dataset is not very convincing. The covered dataset includes image understanding, hallucination detection, and image captioning; none of the mentioned focuses on complex reasoning.

**Essential References Not Discussed:**

Related works are clear

**Experimental Designs Or Analyses:**

See Claims And Evidence

**Methods And Evaluation Criteria:**

The overall method makes sense to me. I agree that the proposed method is a commendable way to address the targeted problem. The only concern about the method is the novelty (see Strengths And Weaknesses section).

The evaluation part is clear and reasonable, but there is potentially an overclaim concern, as mentioned in the Claims And Evidence section.

**Other Comments Or Suggestions:**

None

**Other Strengths And Weaknesses:**

The proposed method is both reasonable and meaningful, with experimental results across multiple datasets demonstrating its effectiveness. However, the novelty of this work is questionable, as the core concept of Reflective Perceptual Learning (RPL) does not introduce fundamentally new learning principles or new findings.

Specifically, the proposed learning framework closely resembles RLHF and self-reflection mechanisms, where a critic model provides iterative feedback to refine predictions. The underlying learning paradigm aligns with process supervision, and the process signal is derived from an LLM-as-a-Judge (plus rule-base scoring) approach. Additionally, the data construction strategy is similar to self-learning or self-distillation techniques. As a result, the overall framework does not present a fundamentally novel methodological contribution beyond its application to vision-language models.

That being said, expanding the success of language model training to VLM represents a meaningful step toward improving real-world applications. Given its potential impact, I am inclined to assign a positive score at this stage.

**Questions For Authors:**

(Refer to the Claims And Evidence Section for more context on the following two questions.)


1. While I appreciate and generally agree with the findings presented in Section 3.1, the evidence provided does not sufficiently demonstrate alignment with human perception. I would like to know if the authors have additional evidence to support this claim or if they would consider narrowing the scope of this claim for greater accuracy. To support this discussion, I am curious about the definition of "ground-truth human attention" in line 243. Is it from human annotation?

2. I believe including datasets focusing on complex reasoning can greatly improve the soundness of the paper. Depending on the reasoning type of interest, the corresponding dataset may be tested, such as common sense reasoning (ScienceQA), complex multimodal reasoning (MultiModalQA), etc.

**Relation To Broader Scientific Literature:**

See Other Strengths And Weaknesses

**Theoretical Claims:**

No serious flaws found

---

> ### Author Rebuttal · Authors · 2025-04-01
>
> We thank the reviewer for the thoughtful feedback and for recognizing the value of our method and experiments. We carefully address the concerns and clarify potential misunderstandings as follows:
> ## Q1: Alignment with Human Visual Focus
> We address this concern by clearly defining “ground-truth human attention” and conducting human evaluations:
>
> **“Ground-truth human attention”** refers to the qualitative visual focus humans naturally adopt when answering questions—typically prioritizing the main subject over background elements. An attention map that better reflects human perception tends to activate more semantically relevant image tokens and highlight regions aligned with human focus patterns [1].
>
> To further support this claim, we conducted a **human evaluation** to compare the image attention of RePer and LLaVA-1.5.
>  - **Setup**: Six annotators (PhD/Master’s students in CV/NLP) were shown anonymized and shuffled attention maps from both models on 100 randomly sampled test images. They were asked to choose: “Which attention map better reflects your own visual focus if you were to answer this question?” The interface is shown in Re-Fig. 2 at https://reper-vl.github.io/ICML-Rebuttal/.
>  - **Metric**: We report the **win rate**—the percentage of cases where RePer’s map was preferred over LLaVA-1.5’s.
>  - **Result**: Image attention maps from RePer was preferred in **70.27%** of cases over LLaVA-1.5, indicating **stronger alignment with human visual focus**.
>
> We will revise the paper to clarify the definition and more accurately scope this claim based on the supporting evidence.
>
> [1] Huang et al. OPERA: Alleviating Hallucination in Multi-Modal Large Language Models via Over-Trust Penalty and Retrospection-Allocation, CVPR 2024.
>
> ## Q2: Advantages in Complex Reasoning
> We appreciate the reviewer’s suggestion and would like to clarify **a potential misunderstanding: our work focuses on perception-oriented tasks, while complex reasoning is positioned as a future direction**. As noted at the end of the Introduction, we see strong perception as a foundation for advanced reasoning capabilities.
>
> To further explore this potential, we evaluated 13B models on **three reasoning benchmarks**—ScienceQA, MMMU, and AI2D—using VLMEvalKit. As shown below, RePer consistently achieves the highest performance, demonstrating stronger reasoning capabilities and generalization.
>
> | Method  | ScienceQA | MMMU  | AI2D  |
> |-|-|-|-|
> | LLaVA-SFT      | 65.76| 34.67 | 47.67 |
> | LLaVA-RLHF     | 63.85| 34.89 | 44.62 |
> | VOLCANO        | 67.29     | 34.33 | 55.89 |
> | LLaVA-1.5      | 68.43 | 35.40 | 58.84 |
> | RePer          | **69.10** | **36.89** | **59.32** |
>
> ## Q3: Novelty of RePer
> **1. Reflection in LVLMs vs. LLMs (Importance & Uniqueness)**: While reflection has been explored in LLMs, it holds distinct importance in the context of LVLMs. Unlike language models, where input tokens are discrete and semantically stable, visual perception involves high uncertainty. LVLMs are more prone to hallucinating nonexistent objects, misinterpreting visual cues, or overlooking salient regions—challenges that extend beyond typical language ambiguity. Our reflection-based method directly addresses these issues by enabling iterative refinement. Beyond improving output quality, it also enhances human-aligned visual attention and reduced hallucination, underscoring its unique value for multimodal perception.
>
> **2. Dual Methodological Contributions**:
> - First, we propose reflective perception learning to LVLMs. where image-grounded, step-wise corrections provide fine-grained rewards for aligning vision and language. This is fundamentally distinct from language-only feedback and proven highly effective.
> - Second, we introduce a reward-weighted unlikelihood training to reinforce high-quality responses while explicitly penalizing suboptimal ones. This mitigates the response collapse observed in prior works (e.g., RISE, SCoRe), where early-stage answers dominate learning regardless of quality, and enables stronger first-turn performance.
>
> **3. Data Construction innovations**:
> While not extensively emphasized in the main paper, our data construction pipeline introduces key innovations by combining VLM-based and rule-based reward signals to construct high-quality reflective conversations. This dual-reward structure improves the precision and interpretability of supervision, making a clear improvement over prior self-reflection datasets (e.g., RISE, SCoRe). Notably, it also resonates with both RM-based and RM-free RL paradigms.
>
> Finally, regarding the notion of novelty itself, we borrow a perspective from Novelty in Science [1]:
> > *If you hear a good idea, there is a moment of surprise and then, the better it is, the more obvious it may seem. If it is easy to explain and obvious in hindsight, this in no way diminishes the creativity (and novelty) of the idea.*
>
> [1] Black. Novelty in Science. Medium. https://medium.com/@black_51980/novelty-in-science-8f1fd1a0a143

---

> > ### Comment · Reviewer_SX8u · 2025-04-03
> >
> > Thanks for the clarification.
> > The author's responses aligned with my initial understanding of the work; thus, I maintained my original overall recommendation.

---

### Official Review · Reviewer_hDWb · 2025-03-17

**Overall Recommendation:** 3

**Summary:**

This paper proposes RePer, which teaches the VLM to iteratively revise and provide gradually better responses given a strong pre-built critic model.

The algorithm works by first collecting responses of varying quality, using these to construct an iteratively revised dataset, and then employing a “Reflective Unlikelihood Training” method that effectively teaches the model to follow this data.

Results show some improvement across benchmarks.

**Claims And Evidence:**

1. The authors claim “RePer’s quantifiable improvements in image understanding, captioning precision, and hallucination reduction” compared to vanilla Llava 1.5. However, given that RePer requires an additional, pre-trained critic model, it is unclear whether the comparison with standalone Llava is fair. I think the paper would benefit from a Best-of-N baseline, which compares RePer’s N-step revision with sampling the solution from Llava-1.5 N times and using the critic to pick the best solution.

2. The authors also claim, “The model achieves strong alignment between model attention patterns and human visual focus,” but the only evidence provided is a single example in Figure 4, which is not convincing. Although Figure 6a shows that average image token activations increase more with RePer, this does not necessarily imply a stronger alignment with human visual focus.

**Essential References Not Discussed:**

No.

**Experimental Designs Or Analyses:**

See Claims And Evidence*

**Methods And Evaluation Criteria:**

Yes, the method appears to be well-motivated, and the benchmarks are solid. However, one issue is that this approach assumes the existence of a robust critic model, which might not always be available.
Additionally see Claims And Evidence 1

**Other Comments Or Suggestions:**

The paper could benefit from deeper discussion with prior works, such as RISE.

**Other Strengths And Weaknesses:**

N/A

**Questions For Authors:**

1. How does RePer compare to Best-of-N against the same critic model?
2. The model receives textual feedback from the critic model, while in training data, the next round of response isn't really correlated with the textual feedback, is this correct?

**Relation To Broader Scientific Literature:**

It extends prior success of improving model's performance through revision, as shown in SCoRe and RISE, from math reasoning, text-based setting to vision-language models.

**Theoretical Claims:**

N/A

---

> ### Author Rebuttal · Authors · 2025-04-01
>
> We thank the reviewer for the positive feedback, particularly for recognizing our work as a “well-motivated method” with solid experiments. Below, we carefully address each of your concerns and clarify potential misunderstandings.
> ## Q1: LLaVA-1.5 BoN vs RePer
> We appreciate the suggestion and clarify **a possible misunderstanding: the critic model is not reqired but optional** for RePer during inference. As discussed in Sec. 4.6 (Lines 370–381), RePer already outperforms LLaVA-1.5 in the first turn without any external feedback, demonstrating a **fair comparison** that reflects the effectiveness of our reflective perception training.
>
> To further address the reviewer’s concern, we conducted another fair comparison on DetailCaps-4870 by implementing a **Best-of-N** (N=1,3,5,6) baseline for LLaVA-1.5, where we sample N responses and use the same critic (LLaVA-Critic-7B) to select the best one for computing the final score.
>
> As shown in the table, even with a larger sampling budget, **LLaVA-1.5-13B Best-of-6 still falls short of RePer-13B’s performance using just a single-turn response**. This underscores RePer’s efficiency and effectiveness in answer generation: rather than relying on brute-force sampling, RePer produces strong initial responses through reflective perception learning mechinism, and further enhances them via feedback-driven refinement.
>
> | Model         | #BoN | #Reflection | Capture | Precision | Recall |
> |---------------|:------:|:-------------------:|---------|-----------|--------|
> | LLaVA1.5  |  1    |        -                 | 51.23   | 65.54     | 43.92  |
> | LLaVA1.5  |  3    | -                 | 51.28   | 65.73     | 43.81  |
> | LLaVA1.5  |  5    | -                 | 51.06   | 65.55     | 43.75  |
> | LLaVA1.5  |  6    | -                 | 51.36   | 66.11     | 43.95  |
> | RePer           | -    | 1                 | 54.29        |  **66.15**  | 47.66       |
> | RePer           | -    | 2                 | 54.68        |  65.24   |   48.68     |
> | RePer           | -    | 3                 | **54.73**        |  64.74         | **49.1**       |
>
> We will clarify the fairness of the comparison more explicitly in the revision.
>
> ## Q2: Alignment with Human Visual Focus
> To address this concern, we conducted a human evaluation to further support the claim that RePer’s image attention better aligns with human visual focus.
>  - **Setup**: We recruited 6 annotators (PhD/Master’s students in computer vision and NLP) and randomly sampled 100 images from the test set. For each image, we collected image attention maps (as shown in Fig. 4) generated by RePer and LLaVA-1.5 during inference. The attention maps were anonymized and randomly shuffled before being shown to annotators, who were asked: “Which attention map better reflects your own visual focus if you were to answer this question?” The annotation interface is shown in Re-Fig. 2 on our supplementary website: https://reper-vl.github.io/ICML-Rebuttal/.
>  - **Metric**: We report the **win rate**, defined as the percentage of cases where RePer’s image attention map was preferred over LLaVA-1.5’s by annotators.
>  - **Result**: Image attention maps from RePer was preferred in **70.27%** of cases over LLaVA-1.5, indicating **stronger alignment with human visual focus**.
> ## Q3: Discussion about RISE
> Please refer to **Lines 196–206** for our discussion on the connection and differences with RISE. While both methods leverage iterative supervision, RePer introduces a distinct imitation learning formulation based on **reward-weighted reflective unlikelihood training**. Moreover, it explicitly proposes a reflective perception mechanism aimed at addressing a key limitation of current LVLMs—the unrealistic assumption of perfect initial responses.
>
> ## Q4: Correlation Between Feedback and Next-Round Response
> We clarify that the next-round responses are **indeed correlated with the critic feedback**. As illustrated in Figure 8, during data construction, each successive response in the conversation is selected based on a higher GPT-4o score than the previous one.
> Concretely, the first response hallucinates non-existent objects like “bottle or vase” which the critic points out. The second response removes these errors and correctly identifies the image as a book. In the third turn, the model further corrects the authorship and improves conciseness, directly following the critic’s feedback. This step-by-step correction process demonstrates a strong alignment between the feedback and the revised responses, providing effective supervision for learning from reflection.

---

### Official Review · Reviewer_VmvG · 2025-03-24

**Overall Recommendation:** 4

**Summary:**

The paper proposes a reflective perception framework, named Reflective Perception (RePer), aimed at enhancing the capabilities of large vision-language models (LVLMs). By introducing dual-model interaction between policy and critic models, RePer seeks to enable LVLMs to iteratively refine their visual perceptions, akin to human observation processes. The core idea is to replace single-pass perception with an iterative feedback loop, allowing models to refine their understanding over multiple rounds. The authors also introduce Reflective Perceptual Learning (RPL), a training approach that fortifies the model's intrinsic reflective capabilities via a constructed visual reflection dataset (LLM-assisted) and reflective unlikelihood training. Experimentation demonstrates improvements in image understanding, captioning precision, and hallucination reduction, with RePer achieving alignment between model attention patterns and human visual focus. The methodology promises robustness in tasks requiring complex reasoning and multi-step manipulation.

**Claims And Evidence:**

The paper's claims are reasonably well-supported by experimental evidence.
* (+) The authors assert that RePer improves image understanding and reduces hallucinations, which is backed by quantitative results across several benchmarks (i.e. MMHal-Bench and HallucinBench).
* (+) The experiments demonstrate superior performance in capturing details and aligning model attention with human focus, which is supported by the results on DetailCaps.
* (-) I am not fully convinced of the eval with GAVIE and GAPE: the evals seem to use the same LLMs involved in finetuning data generation? (sec 2.2, step-2, scoring). One may make the "leveraging discrimination -> generation gradient" argument here but that would need to be further supported empirically which the paper doesn't currently do.

**Essential References Not Discussed:**

On the method front, not that I know of.
For evaluation, perhaps mentioning below to contextualize better the evaluation of detailed text<->image alignment.
* Image->Text:
  - CLAIR: https://arxiv.org/abs/2310.12971
  - DOCCI: https://arxiv.org/abs/2404.19753
* Text->Image:
  - Gecko: https://arxiv.org/abs/2404.16820v1
  - DSG: https://arxiv.org/abs/2310.18235

Since the core proposal with RePer is better image understanding on (fact-grounded) granularity, detailed semantic alignment is a natural topic to touch on.

**Experimental Designs Or Analyses:**

The experimental design appears robust, with comprehensive evaluation across multiple well-chosen benchmarks. The authors thoughtfully designed ablation studies to analyze the influence of reflection turns, scoring disparities, and unlikelihood loss weights. These provide strong evidence of RePer’s advantages and the importance of iterative refinement in perception. However, the experiments could be expanded to include more varied real-world scenarios to ensure generalizability. Additionally, again, because of the lack in strong-enough automated evals, including qualitative **human** feedback would strengthen the assessment of how well model improvements align with human expectations and perception nuances.

Overall, the work looks very promising, but slightly incomplete. To echo some of the key points mentioned on the matter of "what would have made a more complete experimentation"
* Efficiency analysis (with the added reflection budget);
* Either empirically ground effectiveness of Discrimination -> Generation gradient with this design (GPT-eval-GPT), or human eval.

**Methods And Evaluation Criteria:**

The methods and evaluation metrics employed are appropriate (but not the strongest) for addressing the challenges in multimodal perception.
* (+) RePer's dual-model architecture—using both policy and critic models in a feedback loop—adequately mimics iterative human perception, which seems fitting for tackling hallucinations and enhancing refined understanding.
* (+) The use of the MMHal-Bench + HallucinBench + DetailCaps combo makes a good case for the improved capabilities with RePer.
* (-) Again, not fully convinced of the results with GAVIE and GAPE unless empirical evidence is provided to show a) absence or minimal bias introduced with gen + eval with the same models; b) effective discrimination -> generation gradient at work (i.e. model judges better than it generates wrt. task at hand).
* (-) Would be informative to have a A/B comparison on the latency & cost, since RePer likely generates a lot more tokens and many overlap across rounds of responses (e.g. see the example figure 9), so an efficiency eval helps understand the practicality aspect.

On additional comment on the negative point above on human evals -- generally the best replacement for GPT-eval-GPT here would be a medium-sized human evaluation. (Reference: see how this Google's text2image evaluation runs a convincing human evaluation)
Not a practical fix now but should the work missed the acceptance this time, this would be the most ideal action before the next sub in my humble opinion.

**Other Comments Or Suggestions:**

Reference typo: GAIVE -> GAVIE (https://arxiv.org/pdf/2306.14565)

**Other Strengths And Weaknesses:**

Not much beyond what I'd discussed above. The paper has a clean presentation with well-versed prose; provides good empirical investigation (despite slightly flawed evaluation).

**Questions For Authors:**

RePer is a great attempt to move beyond the somewhat "zeigeist" of "let's see if we could prompt LLMs this novel way" with a well-executed (what I'd consider as) knowledge distillation investigation: LLM-assisted data curation -> model finetuning.
How much do you think the reliance on massive LLMs (>100B) form a semi-hard/hard barrier for smaller research groups to make breakthroughs on pushing the boundary of better reasoning? If not, what types of large step functions can we expect which are more free from the reliance.

**Relation To Broader Scientific Literature:**

The paper fits within the broader context of enhancing multimodal models by refining perception processes and reducing hallucinations. It draws on established concepts like Chain-of-Thought reasoning and imitates human-like iterative perception, enriching the current strategies used in vision-language models. RePer's approach contrasts with traditional single-pass perception methods by offering a cyclical refinement process. The empirical focus distinguishes it from theoretical studies and aligns with practical applications in multimodal research areas. However, a deeper engagement with similar iterative learning strategies used in adjacent fields, such as curriculum learning or active learning in machine learning, could enrich the discussion about RePer’s novel contributions and possible future extensions.

**Theoretical Claims:**

The paper does not present formal theoretical proofs but does establish a conceptual framework grounded in reinforcement learning and imitation learning principles. The theoretical basis for Reflective Perceptual Learning aligns with existing reinforcement learning methodologies, yet does not delve deeply into formalizing the convergence properties or theoretical guarantees of the iterative reflective perception process. While the conceptual claims are plausible, they remain largely empirical rather than rigorous theoretical assertions.

---

> ### Author Rebuttal · Authors · 2025-04-01
>
> Thank you very much for constructive feedbacks and for recognizing our presentation quality and experimental design.
> We carefully address each of your concerns below.
> ## Q1: Evaluation Reliability
> To address concerns about potential bias from using the same LLM for both data construction and evaluation, we followed the suggested direction and conducted: **(1) a human evaluation** comparing LLM ranking with human preferences, and **(2) GAPE evaluation using alternative LLMs** not involved in data construction. These results collectively show:
> - **Our proposed GAPE evaluation aligns well with human judgment**, indicating minimal bias.
> - **RePer consistently achieves the highest preference**, both in human evaluations and in GAPE scores across different LLM evaluators.
> - The strong alignment confirms **GPT-4o as an effective discriminator**, which provides a “discrimination to generation” gradient that guides the model to learn accurate preferences.
>
> **1. Human Evaluation**
>
> - **Setup**: This human study uses captions generated by three 13B models on 119 GAPE samples. For each image, three anonymized captions were randomly ordered and shown to six annotators (PhD/Master’s students in CV), who ranked them based on the GAPE criteria (Fig. 7). The interface is shown in Re-Fig.1 at https://reper-vl.github.io/ICML-Rebuttal/.
> - **Metric**:
>   - Mean Rank: The average ranking position across all cases.
>   - Top-1 Rate: The percentage of times a model’s caption was ranked best by humans.
> -  **Results**: As shown below, both human evaluation metrics exhibit a consistent model ranking, aligning with the ranking from the GPT-4o-based GAPE benchmark. Notably, RePer is ranked as the top-1 in 63.87% of cases, validating its effectiveness.
> |Model|Mean Rank↓| Top-1↑|GAPE↑|
> |-|-|-|-|
> |LLaVA-1.5| 2.46|15.13%|77.37|
> |Volcano|2.01|35.29%|78.17|
> |RePer|**1.53**|**63.87%**|**82.54**|
>
> **2. Evaluation with Alternative LLMs**
>
> - To further rule out evaluator bias, we replaced GPT-4o with Claude and Gemini for GAPE scoring on 13B models. As shown below, RePer consistently outperforms baselines, reinforcing the reliability of observed improvements.
> |Model|Gemini↑|Claude↑|GPT-4o↑|
> |-|-|-|-|
> |LLaVA-SFT|78.55|74.64|74.88|
> |LLaVA-RLHF|78.89|74.18|75.36|
> |LLaVA-1.5|80.65|75.87|77.37|
> |Volcano|81.59|76.70|78.17|
> |RePer|**83.39** |**78.41**|**82.54**|
> ## Q2: Latency & Cost
> We justify the practicality of RePer from two angles:
> 1. RePer already **outperforms baselines with its initial response** (Tab. 3), achieving better performance under the same inference cost.
> 2. As suggested, we **evaluate RePer’s efficiency** on 100 samples from the DetailCaps, measuring latency (ms/token) and cost (total generation time).
> As shown below, the added cost from extra turns is modest (e.g., +11 ms/token from 1 to 3 turns), and multi-turn refinement remains **optional** based on budget constraints or quality demands.
> |Model|#Turn|Gen. Token|Cost (ms)|Latency (ms/token)|CAPTURE|
> |-|-|-|-|-|-|
> |LLaVA-1.5|1|85.83|4750.0|55.34|51.23|
> |RePer|3|275.42|18377.5|66.73|55.55|
> ## Q3: Reliance on Large-scale LLMs
> We believe breakthroughs are possible without relying on >100B LLMs.
>
> In RePer, core improvements stem from the reflective perception mechanism and RPL paradigm—not from reliance on large-scale LLMs. Both the reward scoring and the critic model can be implemented without large models: Fig. 2 (Step-3) illustrates rule-based scoring aligning vision and text, and Tab. 3 shows a smaller LLaVA-Critic-7B can be an effective critic.
>
> A promising direction beyond large LLMs is to build more reliable and efficient supervision systems. Large models are convenient for data generation but often introduce hallucinations that cap smaller models’ performance. Combining expert models for pre-/post-processing, or incorporating lightweight human-in-the-loop feedback [1] for corretion/validation, offers a cost-effective way to improve supervision quality.
>
> Another direction is to pursue efficient modeling paradigms. Recent work such as OpenAI-O1/Deepseek-R1 employ RL methods like PPO/GRPO on verified data, enabling models to better exploit internal capabilities and self-generate high-quality reasoning trajectories.
>
> [1] Garg R et al. (2024). Imageinwords: Unlocking hyper-detailed image descriptions. Google.
> ## Q4: Relations to Iterative Learning Strategies
> RePer shares conceptual ties with active and curriculum learning: it uses feedback to refine predictions and progressively improves responses from coarse to accurate. Unlike traditional approaches, this progression is self-structured within the RePer’s own decision loop.
> As a future direction, integrating uncertainty-based reflection or organizing learning trajectories from simple to complex corrections could further enhance adaptability.
> ## Q5: Reference & Typo
> Thank you for the helpful references. We carefully considered them for designing our human evaluation, and will include the citations and fix the typo in the revision.

---

### Decision · Program_Chairs · 2025-05-01

**Decision:**

Accept (poster)

**Comment:**

This paper proposes a multi-step framework for visual perception based on a perception-feedback loop. Implementing such a system is challenging because using a model to self-correct it's own outputs has often been shown empirically to be ineffective.

Here, the authors begin by collecting a dataset of multi-turn reflective dialogs, by first generating initial responses to a question, then scoring these responses with both model-based and rule-based approaches, before finally ordering the responses from worst-to-best into a dialog (steadily improving responses interspersed with feedback).

Experiments are conducted finetuning finetuning LLaVA-1.5 on the collected dataset using Reflective Unlikelihood Training (RUT), a variation of unlikelihood training. Experiments show consistent improvements on image understanding, hallucination detection and detailed image captioning.

All reviewers voted to accept the paper and I agree. However, I found some of the emphasis in the paper (for example, on the perception-feedback loop) to be misaligned with the relative importance of the paper's contributions. According to Table 3, it seems like most of the benefit to the approach comes from the finetuning process, and not the multi-round perception-feedback loop. Further, the critic model is from a different paper and so isn't a contribution.

However, to my knowledge the RUT loss is novel. The same sequence appears in both the likelihood and the unlikelihood term, but with weightings determined by normalized reward. The advantage over weighted supervised finetuning appears to be that the unlikelihood term introduces explicit repulsion from bad responses. The author's acknowledge that "RPL is a Free-Form Preference Optimization" (Section 3.2). Personally, I think RUT begs for more time spent on the direct comparison to DPO (the collected dataset can be interpreted in terms of pairwise preferences). This is included in Table 4 but with insufficient details of the implementation.

Overall, this is a solid paper. I encourage the authors to consider all feedback including tweaking the focus more towards reward-weighted reflective unlikelihood training (e.g. perhaps greater mention in the title/abstract, and improved details around Table 4).